# *Lactobacillus gasseri* ATCC33323 affects the intestinal mucosal barrier to ameliorate DSS-induced colitis through the NR1I3-mediated regulation of E-cadherin

**Guanru Qian**[1,2☯], **Hui Zang**[1,2☯], **Jingtong Tang**[1,2], **Hao Zhang**[1,2], **Jiankang Yu**[1,2], **Huibiao Jia**[1,2], **Xinzhuang Zhang**[1,2], **Jianping Zhou**[1,2] *

1 Department of Gastrointestinal Surgery & Hernia and Abdominal Wall Surgery, the First Hospital, China Medical University, Shenyang, China, 2 Department of Shenyang Medical Nutrition Clinical Medical Research Center, Shenyang, China

☯ These authors contributed equally to this work.
* zjphama@163.com

**Data Availability Statement:** The authors confirm that all data underlying the findings are fully available without restriction. All relevant data are

## Abstract

Inflammatory bowel disease (IBD) is an immune system disorder primarily characterized by colitis, the exact etiology of which remains unclear. Traditional treatment approaches currently yield limited efficacy and are associated with significant side effects. Extensive research has indicated the potent therapeutic effects of probiotics, particularly *Lactobacillus* strains, in managing colitis. However, the mechanisms through which *Lactobacillus* strains ameliorate colitis require further exploration. In our study, we selected *Lactobacillus gasseri* ATCC33323 from the intestinal microbiota to elucidate the specific mechanisms involved in modulation of colitis. Experimental findings in a DSS-induced colitis mouse model revealed that *L. gasseri* ATCC33323 significantly improved physiological damage in colitic mice, reduced the severity of colonic inflammation, decreased the production of inflammatory factors, and preserved the integrity of the intestinal epithelial structure and function. It also maintained the expression and localization of adhesive proteins while improving intestinal barrier permeability and restoring dysbiosis in the gut microbiota. E-cadherin, a critical adhesive protein, plays a pivotal role in this protective mechanism. Knocking down E-cadherin expression within the mouse intestinal tract significantly attenuated the ability of *L. gasseri* ATCC33323 to regulate colitis, thus confirming its protective role through E-cadherin. Finally, transcriptional analysis and *in vitro* experiments revealed that *L. gasseri* ATCC33323 regulates *CDH1* transcription by affecting NR1I3, thereby promoting E-cadherin expression. These findings contribute to a better understanding of the specific mechanisms by which *Lactobacillus* strains alleviate colitis, offering new insights for the potential use of *L. gasseri* as an alternative therapy for IBD, particularly in dietary supplementation.

within the paper and its supporting information files.

**Funding:** JZ was supported by the Applied Basic Research Program from the Science and Technology Agency of Liaoning Province, China (2022JH/101300025). The funders did not play any role in the study design, data collection and analysis, decision to publish, or preparation of the manuscript.

**Competing interests:** The authors have declared that no competing interests exist.

## Author summary

With increasing interest in the efficacy of probiotics to improve IBD, we set our sights on *Lactobacillus*, which has great potential. *Lactobacillus gasseri* ATCC33323 has been shown to have definitive efficacy in gastritis, but has not been definitively reported in colitis. Here, we first found an amelioration of DSS-induced colitis in mice by gavage with *Lactobacillus gasseri* ATCC33323. At this stage, little research has been done on the mechanisms by which *Lactobacillus* improves IBD, based on the establishment of transgenic mice with semiknockout of E-cadherin in the intestine, we have clarified that *Lactobacillus gasseri* ATCC33323 is targeting E-cadherin to exert a therapeutic effect on colitis in mice. This is also the first time that a mouse model of E-cadherin semiknockout in the intestine has been established. Moreover, through in vitro experiments we found that *Lactobacillus gasseri* ATCC33323 plays a role in regulating E-cadherin through NR1I3. Our study provides a new perspective on improving IBD through microorganisms.

## 1. Introduction

IBD includes ulcerative colitis (UC) and Crohn's disease (CD), and the causes of IBD are currently unclear. Although conventional and classic treatment plans can be used for the treatment and prevention of IBD at present, the effectiveness of these treatments is limited, and they can cause serious adverse reactions and incur high costs [1]. Therefore, researchers have begun to focus on probiotics, which have shown strong efficacy in mouse models of intestinal inflammation and have been found to influence clinical effects in patients with UC [2,3]. The gut microbiota plays a crucial role in the regulation of the immune system and the pathogenesis of intestinal inflammation [4]. Interventions involving the gut microbiota, probiotics, or diet can restore the balance of interactions between the gut microbiota and immune ecology to reduce inflammation and damage in the intestine, which has been shown by previous experiments[5, 6].

The intestinal barrier is composed of very complex junctions between cells, including tight junctions (TJs), adherens junctions (AJs), desmosomes, and gap junctions, which can isolate the body's internal environment from the contents of the gut[7]. TJs are highly dynamic structures near the apical surface of intestinal cells and are composed of several transmembrane proteins, such as occludin, claudins, junction adhesion molecules, and the zona occludens (ZO) family [8]. AJs are located downstream of TJs. AJs contain abundant E-cadherin and can regulate intercellular adhesion. The p120-catenin, α-catenin, and β-catenin proteins are composed of repeatedly folded characteristic amino acids that can bind to the cytoplasmic portion of E-cadherin within cells.

Numerous studies have shown that the use of *Lactobacillus* may be an alternative method for treating IBD, as several strains of the *Lactobacillus* genus have been shown to regulate cytokine activity, improve the integrity of the intestinal barrier, and affect proteins such as ZO-1, occludin, claudin1, β-catenin, and E-cadherin. These strains can also increase the secretion of MUC2 in intestinal epithelial cells, reduce intestinal epithelial permeability, and protect intestinal barrier function [9, 10].

*Lactobacillus gasseri* ATCC33323 is a human-derived bacterial strain extracted from the human body and planted in the digestive tract and vagina of young people and adults [11]. It is a native probiotic strain in the digestive tract of humans and animals [12]. Beneficial effects, such as anti-inflammatory effects, oxidative stress regulation, and *Helicobacter pylori* inhibition, have been reported for some bacterial strains of *Lactobacillus gasseri* in the human body

[13–15]. Therefore, *Lactobacillus gasseri* can be considered a candidate strain for probiotics. Stephanie Hummel et al. reported that after *Lactobacillus gasseri* is cultured with T84 cells, the expression of E-cadherin increases at the mRNA and protein levels so that AJs can be stabilized and the barrier function of gastrointestinal epithelial cells can be enhanced, reducing their permeability [16].

The constitutive androstane receptor (CAR, NR1I3) is a member of the nuclear receptor superfamily. NR1I3 is a well-known transcription factor that can regulate the expression of many genes involved in the metabolism and transportation of exogenous and endogenous chemicals [17]. The gut microbiota can affect the expression of NR1I3 target genes, the receptors themselves, and the activity of NR1I3[18].

The aim of this study was to evaluate the role of the human native bacterial strain *L. gasseri* ATCC33323 in a DSS-induced murine model of colitis and to apply mouse hybridization to construct transgenic mice with semiknockout of the intestinal *Cdh1* gene. We selected the semiknockout mouse model of E-cadherin because the mice cannot survive after full knockout of E-cadherin in the intestine. To determine the mechanism of action of *L. gasseri* ATCC33323 in colitis and the effects of intestinal bacteria, the possibility that NR1I3 may act as a mechanism through which *L. gasseri* ATCC33323 regulates E-cadherin was also explored via RNA-seq and *in vitro* cellular experiments.

## 2. Materials and methods

### 2.1. Ethics statement

This study was approved by the Animal Ethics Committee of ChinaMedical University (ethical approval code: KT2023176).

### 2.2. Bacterial strains and culture conditions

*Lactobacillus gasseri* ATCC33323 was obtained from Zhili Zhongte Biological Technology Co., Ltd. (Wuhan, China). First, *L. gasseri* ATCC33323 was inoculated into MRS liquid medium (Hopebio, China). The mixture was subsequently incubated on a constant temperature shaker at 37°C for 48 hours, after which the bacteria were collected via centrifugation at $3500 \times g$ and 4°C for 10 minutes. The cells were then washed 3 times with PBS and suspended in PBS. After the bacterial density was adjusted on the basis of the counting results, bacteria at densities of $1 \times 10^{10}$ cfu/mL and $1 \times 10^{8}$ cfu/mL were suspended in PBS to prepare bacterial suspensions, and the bacterial suspensions were stored at 4°C until use or frozen at -80°C.

### 2.3. Animals and experimental design

The mice used in this study were C57BL/6 mice, which included homozygous E-cadherin allele *Cdh1^{flox/flox}* mice and *Villin-Cre* transgenic tool mice acquired from Cyagen Biosciences Co., Ltd. *Cdh1^{flox/flox}* mice were crossed with *Villin-Cre* mice and genotyped via PCR to produce *Villin-Cre;Cdh1^{wt/flox}* mice and *Cdh1^{wt/flox}* mice and *Cdh1^{wt/flox}* littermates were used as WT controls for *Villin-Cre;Cdh1^{wt/flox}* mice. All experiments were performed using littermate animals from breeding pairs. The 24 *Villin-Cre;Cdh1^{wt/flox}* mice aged 6–8 weeks were divided into 3 groups, which were named the CDH1+/- PBS, CDH1+/- DSS, and CDH1+/- LAB groups, with 8 mice in each group. Moreover, 24 WT (*Cdh1^{wt/flox}*) mice aged 6–8 weeks were divided into 3 groups, which were named the NC PBS, NC DSS, and NC LAB groups, with 8 mice in each group. During the experiment, the mice in the CDH1+/- LAB group and the NC LAB group were gavaged daily at a density of $1 \times 10^{10}$ cfu/mL and with 200 μL/d *L. gasseri* ATCC33323 for 21 consecutive days. At the same time, 200 μL/d PBS was administered to the

mice in the CDH1+/- PBS, CDH1+/- DSS, NC PBS, and NC DSS groups. During the 15–21 day period, the sterile drinking water of the CDH1+/- DSS, CDH1+/- LAB, NC DSS, and NC LAB groups was replaced with 2.5% (w/v) DSS (36,000–50,000 Da, MP Bio, Canada) to induce colitis. All the mice were acclimated to the laboratory environment of China Medical University for at least one week, after which the temperature (20°C±2°C), humidity (50%±5%), and light–dark cycle was maintained. The mice were raised in ventilated cages where sterile water and normal mouse feed were provided.

## 2.4. Mouse genotype identification

The tail was collected 2–3 mm from the end of each mouse, after which lysis buffer and balance buffer were added. The PCR mixture was subsequently prepared, and the most suitable conditions for DNA amplification were selected according to the instructions of the genotyping kit (APExBIO, USA). The PCR products were put into a 2% agarose gel, agarose gel electrophoresis was performed with 1×TAE, and images were captured by a gel imager.

The sequences of the primers used were as follows: Villin-Cre: F, GTGGTTTGGTTTGGTTT CCTCTGCATAAGA; R, GCAGGCAAATTTTGGTGTACGGTCA; target, 567 bp.

flox/flox *Cdh1*: F: GGGTCTCACCGTAGTCCTCA; R: GATCTTTGGGAGAGCAGTCG, Mutant = 310 bp, Wild type = 243 bp.

## 2.5. Cell lines and culture

Caco2 cells, HCT116 cells (human colorectal cancer epithelial cells), and human embryonic kidney 293T cells were obtained from the cell bank of the Chinese Academy of Sciences (Shanghai, China). Caco2 cells were cultured with the addition of 20% fetal bovine serum (FBS; HyClone, Logan, UT, USA), 100 U/mL penicillin, and 100 µg/mL streptomycin in minimum essential medium (MEM) (Priscilla, China). HCT116 cells and HEK293T cells were cultured with the addition of 20% fetal bovine serum, 100 U/mL penicillin and 100 µg/mL streptomycin in Dulbecco's modified Eagle's medium (MEM). The cell culture environment was humidified at 37°C and 5% $CO_2$. The culture medium was changed and checked to ensure proper cell morphology.

## 2.6. Cell coculture and transfections

The cells were inoculated at a density of $1\times10^5$ cells/well into a 6-well plate, and $1\times10^8$ cfu/mL *L. gasseri* ATCC33323 (supplemented with *Lactobacillus* at an MOI of 1:100) and culture medium without antibiotics were added after the cells reached 80%-90% confluence. The plates were cultivated in an incubator at 37°C and 5% $CO_2$ for a total of 3, 6, or 12 hours.

NR1I3 siRNA and its corresponding negative control were synthesized by GenePharma (Shanghai, China). Colon cancer cell lines in which NR1I3 was silenced were generated via Lipofectamine 3000 (Invitrogen, Carlsbad, CA, USA) according to the manufacturer's instructions. The two specific interference sequences used were as follows: NR1I3-homo427: F: GCAUGAGGAAAGACAUGAUTT; R: AUCAUGUCUUUCCUCAUGCTT; NR1I3-homo981: F: GAG CCUGAGUAUGUGCUCUTT; R: AGAGCACAUACUCAGGCUCTT.

## 2.7. Inflammatory response of Caco2 and HCT116 cells stimulated with LPS to *L. gasseri* ATCC33323

To assess their anti-inflammatory activity, we used medium-treated Caco2 and HCT116 cells as negative controls. The penicillin–streptomycin in the culture medium was removed when the adherent cells reached 70%-80% confluence. Caco2 and HCT116 cells were incubated with

medium containing $1\times10^8$ cfu/mL *L. gasseri* ATCC33323 for 12 h. Cells without *L. gasseri* ATCC33323 were used as the positive control group. The bacteria in the medium were then removed from the 6-well culture plates. After being stimulated with or without *L. gasseri* ATCC33323, Caco2 and HCT116 cells were incubated with 1 μg/mL LPS from *E. coli* 0111:B4 (Sigma, St. Louis, MO) for another 4 h[19]. After incubation, the cells were washed three times with sterile PBS and extracted for the next step.

## 2.8. Disease activity index (DAI)

During the DSS modeling period, the body weight, fecal shape, and blood stool status of the mice were measured daily and observed. The fecal occult blood test was performed via the o-toluidine method according to the instructions of the reagent kit (Baiolaibo, China). The scoring criteria for weight loss were as follows: 0 (within 1% of baseline), 1 (decrease from 1% to 5%), 2 (decrease from 5% to 10%), 3 (decrease from 10% to 15%), or 4 (decrease >15%). Stool consistency: 0 = normal; 2 = soft, with normal morphology (loose stools that do not stick to the anus); 4 = loss of form/diarrhea (liquid stools adhering to the anus). Stool bleeding status was scored as 0 = negative, 2 = detectable blood with a colorectal detection kit, and 4 = gross. The DAI (body weight+stool consistency+blood stool) is the criterion for assessing the severity of colitis in mice[20].

## 2.9. Assessment of MPO, SOD, and MDA levels

The contents of SOD (WST-1 method), MDA (TBA method), and MPO (colorimetry) were determined according to the instructions of the reagent kit (Nanjing Jiancheng Biotechnology Research Institute) after the colon tissue was weighed and the tissue homogenate was prepared. The expression levels of SOD, MDA, and MPO were determined as U/mg prot, nmol/mg prot, and U/g, respectively.

## 2.10. *In vivo* intestinal permeability assay

The mice were deprived of food and water for 12 hours after DSS treatment. Then, the mice were gavaged with FITC-dextran (4,000 Da, Sigma Aldrich) at doses of 0.5 mg/g and 50 mg/ml, and the serum was collected after 4 hours. The FITC concentration in the serum was detected by using a microplate reader (excitation: 485 nm; emission: 528 nm), and intestinal permeability was evaluated by using the FITC density, which was calculated on the basis of the plotted criteria curve.

## 2.11. Hematoxylin–eosin (H&E) staining

Fresh colon tissue was removed from the sacrificed mice, and was placed in 4% paraformaldehyde for 24 hours. The tissues were sliced into 3–5 micron sections after they were sealed with paraffin. After heating at a constant temperature of 60˚C for 2 hours, the samples were hydrated and deparaffinized. The sections were stained with hematoxylin and eosin. A 200x field of view was observed under a microscope (Nikon E100, Japan) to obtain slice images, the pathological formation of colon tissue was observed, and histological scoring was subsequently performed on the basis of the criteria in Table 1 [21].

## 2.12. Immunohistochemical staining and immunofluorescence staining

After deparaffinization, hydration, and antigen retrieval, the mouse colon tissue sections were removed, endogenous peroxidases were blocked, and the sections were treated with an immunohistochemical kit (Maixin, China). The sections were incubated with HRP-conjugated

**Table 1. Histopathology grading system.**

| Score | Inflammatory infiltrate | Extent of inflammation | Crypt damage |
|---|---|---|---|
| 0 | None | None | None |
| 1 | Mild | Mucosa | 1/3 damaged |
| 2 | Moderate | Mucosa and submucosa | 2/3 damaged |
| 3 | Severe | Transmural | Crypt loss but surface epithelium present |
| 4 | / | / | Both crypt and surface epithelium loss |

secondary antibodies against ZO-1 (1:200; Proteintech, USA), E-cadherin (1:500; Proteintech, USA), β-catenin (1:500; Proteintech, USA), p120-catenin (1:200; Proteintech, USA), MUC2 (1,500; Proteintech, USA), occludin (1:200; Proteintech, USA), and claudin-1 (1,200; Proteintech, USA) overnight. Then, the cell nuclei were stained with hematoxylin, and whole sections were stained with DAB. When observed by microscope at a magnification of 200×, the nuclei were stained blue with hematoxylin, and DAB-positive cells were defined as brown/yellow. The staining scores were 1 (0–25%), 2 (25–50%), 3 (50–75%), and 4 (75–100%) [0 (negative), 1 (weak), 2 (moderate), and 3 (strong)] on the basis of the staining range and intensity. The final IHC pathological score was determined by three pathologists from our hospital, and the final staining score (0–7) was obtained according to the above criteria.

The tissue sections were blocked with 5% BSA for 30 minutes after antigen retrieval, after which the tissues were incubated with a β-catenin rabbit antibody (1:200, Proteintech, USA), a p120-catenin rabbit antibody (1:50, Proteintech, USA), and an E-cadherin mouse antibody (1:150, Abcam, USA). The sections were stained with FITC-labeled IgG (1:20 dilution) and DAPI (Sigma, St. Louis, MO, USA), and images were obtained via a fluorescence microscope (Nikon Microphot-FX, Tokyo, Japan) after resin sealing. The average fluorescence intensity was acquired via ImageJ.

## 2.13. Protein extraction and western blot analysis

Total protein was obtained from mouse colon tissue and colon cancer cell samples. A BCA protein assay kit (BOSTER, China) was used to determine the total protein concentration. The samples were subsequently separated via 10% SDS polyacrylamide gel electrophoresis, after which the proteins were transferred to polyvinylidene fluoride (PVDF) membranes. The PVDF membrane was blocked with 5% skim milk and incubated with antibodies in a shaker at 4°C overnight. The types and dilution ratios of the antibodies used were as follows: ZO-1 (1:500; Proteintech, USA), E-cadherin (1:2000; Proteintech, USA), claudin1 (1:500; Proteintech, USA), β-actin (1:1000; Proteintech, USA), p120-catenin (1:1000; Proteintech, USA), β-catenin (1:1000; Proteintech, USA), occludin (1:500; Proteintech, USA), and NR1I3 (1:1000; Proteintech, USA). Afterward, the membrane was washed three times and incubated with HRP-conjugated goat anti-rabbit and goat anti-mouse secondary antibodies at room temperature for 2 h (1:10000; BOSTER), after which an enhanced chemiluminescence (ECL) substrate kit (Thermol Biotech, USA) was used to detect the signal in the luminescent membrane. ImageJ software was subsequently used to quantify the bands through grayscale value analysis, with β-actin serving as an internal parameter.

## 2.14. Real-time quantitative polymerase chain reaction (qPCR)

Total RNA was extracted from colon tissue and colon cancer cell lines using TRIzol reagent (Takara, Japan) and was reverse transcribed into cDNA by using a reverse transcription mixed reagent kit (Takara, Japan). Then, IL-1β, IL-6, and TNFα were detected with β-actin as an internal parameter after qPCR was conducted by using GoTaq SYBR-Green qPCR Master Mix (Takara, Japan). The polymerase chain reaction (PCR) and amplification conditions were the optimal conditions recommended in the manufacturer's manual. A melting curve was generated after cDNA was subjected to qPCR, and each pair of primers had only one peak and one product. CT was used as the cutoff, and the $2^{-\Delta\Delta Ct}$ method was used to calculate the relative expression. The sequences of the primers used are described in S1 Table.

## 2.15. Mouse fecal genome extraction and analysis

After collection, total DNA was extracted from mouse stool via the hexadecyl trimethyl ammonium bromide (CTAB) method, after which the DNA extraction quality was assessed via agarose gel electrophoresis. Simultaneously, a UV–visible spectrophotometer was used to quantify the DNA. Afterward, PCR amplification of the V3-V4 region of the bacterial gene was performed by using the forward primer 341F (5'-CCTACGGGNGGCWGCAG-3') and the reverse primer 805R (5'-GACTACHVGGGTATCTAATCC-3'). The PCR amplification products were detected via 2% agarose gel electrophoresis and recovered with an AMPure XT bead recovery kit. The next step involved the use of a NovaSeq 6000 sequencer for $2 \times 250$ bp double-ended sequencing, and the sequencing data were processed via the QIIME2 version of the Microbial Ecology Quantitative Analysis System. Furthermore, an OTU table was generated to record the abundance of each OTU and classify all the OTUs in each sample. System clustering of the different species ratios was performed via principal coordinate analysis (PCoA), nonmetric multidimensional scaling analysis (NMDS), and the nonweighted arithmetic average pairing method (UPGMA) to determine the species ratio in each sample.

## 2.16. DNA copy numbers of *Lactobacillus spp*. and *L. gasseri* ATCC33323 determined via real-time quantification PCR

The extracted genomic DNA of *Lactobacillus* was used as a template to amplify the 231-bp feature fragment of the *Lactobacillus spp*. gene and the 322-bp feature fragment of the *L. gasseri* ATCC33323 gene, which were cloned and inserted into the pMD-19T vector. The recombinant plasmids were sequenced and identified, and then used as standards with tenfold gradient dilutions to carry out RT–qPCR via the SYBR fluorescent dye method and establish a standard curve[22].

DNA was isolated from a collected stool sample via a magnetic stool DNA kit (DP712, TIANGEN, China). An Applied Biosystems 7500 Fast real-time system (Thermo Fisher, USA) and SYBR Premix Ex Taq II (Takara, Japan) were used to perform qPCR. The specific sequences of primers used were as follows: *Lactobacillus spp*.: F: TGGAAACAGRTGCTAATAC CG; R: GTCCATTGTGGAAGATTCCC[23]; *L. gasseri* ATCC33323: F: TGGAAACAGRTGCT AATACCG; R: CAGTTACTACCTCTATCTTTCTTCACTAC[24]. The reaction mixture (20 μL) included SYBR Premix Ex Taq II (10 μL), forward and reverse primers (0.8 μL), ROX Reference Dye II (0.4 μL), sterile deionized water (6 μL), and the DNA template (2 μL). qPCR was performed as follows: 95˚C for 30 min, 40 cycles of 95˚C for 5 s and annealing at optimal temperatures for 34 s at 60˚C. The specificity of the qPCR primers was regulated by generating melting curves. Absolute quantification was used to determine the number of bacteria expressed per gram of feces via qPCR. The data are presented as the mean values of duplicate

qPCR analyses. The qPCR products were separated via agarose gel electrophoresis and photographed.

## 2.17. RNA extraction, library construction and bioinformatics analysis

First, Dynabeads Oligo(dT) (Thermo Fisher, USA) were used to specifically capture mRNAs that contained PolyA by two rounds of purification after separating and purifying the RNA from the mouse intestine. After the mRNA was fragmented, it was then synthesized into cDNA. The fastp package (https://github.com/OpenGene/fastp) was used to verify the sequence quality after the reads that contained adaptor contamination, low-quality bases, and undetermined bases with default parameters were removed. The next step was to use gffcompare (https://github.com/gpertea/gffcompare/) to reconstruct a comprehensive transcriptome after merging all the transcriptomes from all the samples. Once the final transcriptome was generated, StringTie was used to estimate the expression levels of all the transcripts, and the expression levels of the RNAs were determined by FPKM (FPKM = [total_exon_fragments/ mapped_reads(millions) × exon_length(kB)]. Finally, the differentially expressed RNAs with a fold change > 2 or < 0.5 were selected, and a parametric F test was used to compare the results with those of nested linear models (p value < 0.05) via the R package edgeR.

## 2.18 Chromatin immunoprecipitation

Initially, 37% fresh formaldehyde was used to crosslink the Caco2 colon cancer cell lines and HCT116 colon cancer cell lines in a 15-cm culture dish, after which the crosslinking process was terminated after 10 minutes with glycine. PBS supplemented with protease inhibitors was used to collect the compounds. Afterward, the chromatin DNA of the sample was sheared into fragments of 250–800 bases, and the DNA fragments bound to NR1I3 were aggregated by immunoprecipitation after the addition of 5 μl of anti-NR1I3 antibody (CST, USA) and 2 μl of IgG antibody (Santa Cruz, Japan). Then, primers were designed on the basis of potential binding sites, and the *CDH1* promoter subregion containing the NR1I3 binding site was analyzed via PCR for DNA amplification. The PCR products were separated via agarose gel electrophoresis and photographed. The sequences of the primers used were F: CTGACTCACTAACCCA TGAAGCand R: AGAGGGGATCTCACTATGTTGC.

## 2.19. Dual-luciferase reporter assay

GeneChemC was used to construct a luciferase reporter plasmid (*CDH1*-wt plasmid, *CDH1*-mt plasmid, NR1I3 plasmid, and pGMLR-TK luciferase reporter plasmid), after which HEK293T cells were inoculated into a 24-well plate and cultured overnight. The next step involved cotransfecting the NR1I3 plasmid, pGMLR-TK luciferase reporter plasmid, and the *CDH1* promoter-wt plasmid in HEK293T cells and cotransfecting the first two plasmids and the *CDH1* promoter mut plasmid in HEK293T cells. After transfection for 48 hours, the cells were harvested according to the instructions of the Luciferase Reporter Gene Kit (Beyotime Biotechnology, China), and their luciferase activities were determined via the Varioskan Flash System (Thermo Fisher).

## 2.20. Statistical analysis

All the experiments were conducted in triplicate, and the data were statistically analyzed and plotted via GraphPad Prism (USA) software; the data are expressed as the means ± standard deviations (SDs). A t test and one-way analysis of variance (ANOVA) were used for statistical analysis. The statistical significance is expressed as *, P<0.05; **, P<0.01; and ***, P<0.001,

which represent significant, strongly significant, and extremely significant differences, respectively.

## 3. Results

### 3.1. E-cadherin is involved in the attenuation of DSS-induced physiological lesions in mice by *L. gasseri* ATCC33323

To investigate the regulatory effect of *L. gasseri* on E-cadherin in the intestine, hybridized homozygous C57BL/6 mice harboring two floxed *Cdh1* alleles and C57BL/6 mice harboring the *Villin-Cre* recombinase enzyme were used to synthesize mice that carried one wild-type *Cdh1* allele, one floxed *Cdh1* allele and possibly Cre transgenic genes (Fig 1A). The heterozygous *Villin-Cre;Cdh1^{wt/flox}* mice were screened via agarose gel electrophoresis (Fig 1B), and E-cadherin was specifically deactivated in the developing intestinal tract[25].

Since lactobacilli can lead to weight and fat loss[26], we focused on evaluation of the percentage of weight loss. There was a significant difference in body weight between the NC DSS group and the NC LAB group on the sixth day of modeling (P<0.05). The body weights of the mice in the CDH+/- LAB group were also lower than those in the NC LAB group on the sixth day (P<0.05) (Fig 1E).

The DAI was used to score the mice on the basis of their body weight, fecal shape, and presence of bloody stools, and a higher score indicates more severe colitis [27]. On the fifth day after DSS induction, the DAI was greater in the CDH1+/- DSS group than in the NC DSS group (P<0.05), demonstrating that the absence of E-cadherin in the intestine accelerated the progression of colitis. The DAI scores were greater in the NC DSS group than in the NC LAB group beginning on the second day (P<0.05). Symptoms of colitis, such as weight loss or sparse feces, appeared in the NC DSS group, whereas the mice in the NC LAB group presented no symptoms at the same time. The DAI scores of the NC LAB group and CDH1+/- LAB group were significantly different on the fourth day (P<0.05) (Fig 1C). During the DSS modeling period, the mice in the CDH1+/- DSS group presented early colitis symptoms and died early, while the mice in the NC LAB group presented the highest survival rate (Fig 1D).

### 3.2. E-cadherin is participating in the alleviation of DSS-induced pathological lesions and intestinal permeability changes in mice by *L. gasseri* ATCC33323

As shown in Fig 1H, the colon lengths of the CDH1+/- DSS group and NC DSS group were significantly shorter than those of the other groups; additionally, the intestinal wall was congested and severely adhered to the surrounding tissues, and bloody contents were visible throughout the colon. In the NC LAB group, there was no significant shortening of colon length, and no congestion or hematoma was observed. The improvement in colon length was significantly reduced, and intestinal wall congestion was visible in the CDH1+/- LAB group. The colon lengths of all the mice were measured at the same time, and the results demonstrated that the colon length of the mice in the NC LAB group was longer than that of both the CDH1+/- LAB group (P<0.001) and the NC DSS group (P<0.001); moreover, there was no significant difference in the CDH1+/- LAB group compared with the CDH1+/- DSS group (Fig 1F).

The levels of FITC-dextran in mouse serum were measured to evaluate the permeability of the intestine. The results revealed that the lowest intestinal permeability of the mice was observed in the NC PBS and CDH1+/- PBS groups, and the intestinal permeability of the mice in the NC LAB group was lower than that of the mice in the NC DSS group (P<0.05) and the

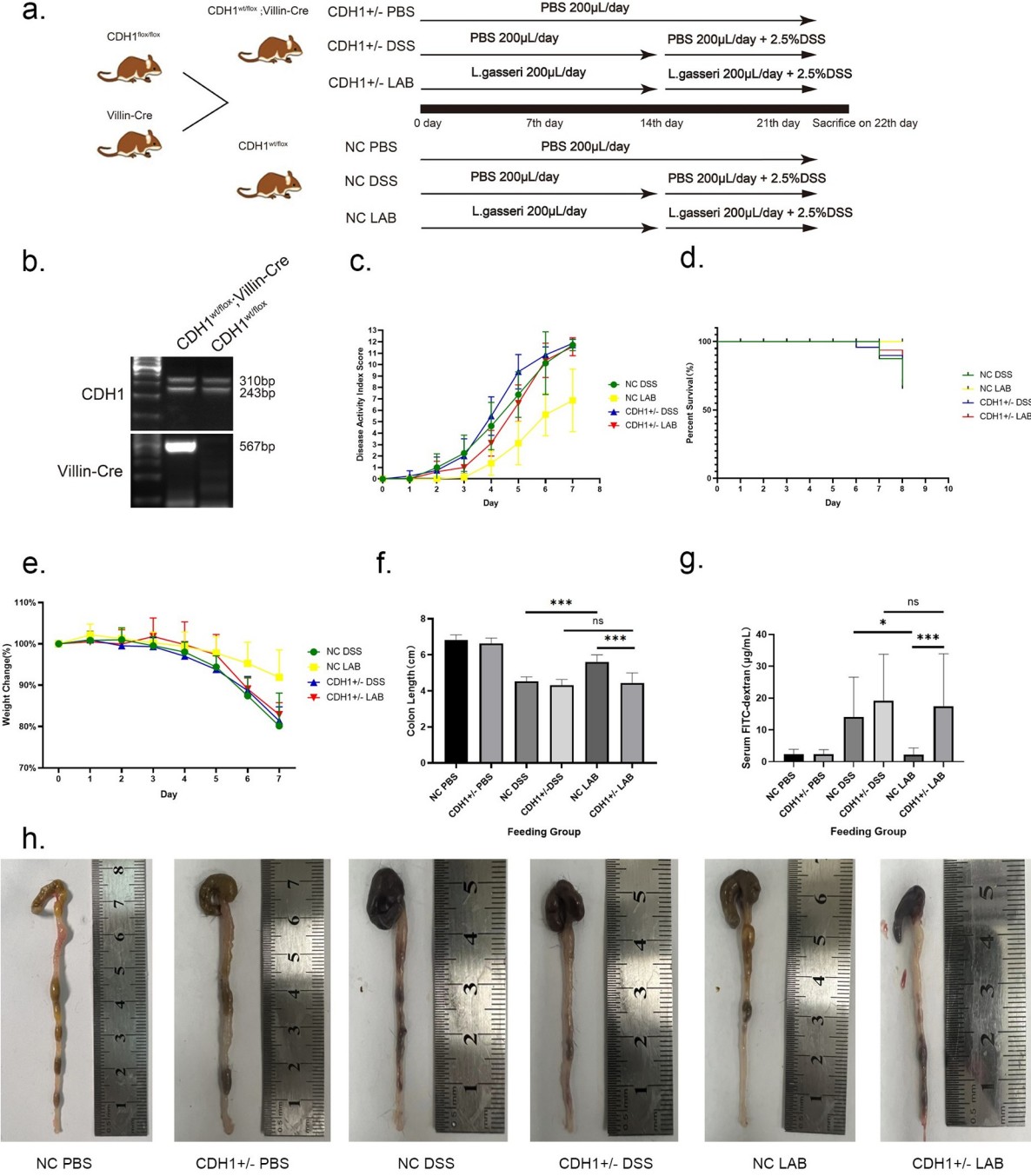

**Fig 1. Effects of *L. gasseri* ATCC33323 supplementation on DSS-induced colitis symptoms.** a. Study design. Villin-Cre;CDH1[wt/flox] mice were crossbred to serve as the experimental group, and CDH1[wt/flox] mice served as the control group. *L. gasseri* ATCC33323 or PBS was gavaged for 21 consecutive days, with simultaneous ad libitum access to 2.5% (w/v) DSS in the latter 7 days. b. PCR analysis of hybrid offspring between CDH1[flox/flox] mice and transgenic mice harboring Villin-Cre;CDH1: Mutant = 310 bp, Wild = 243 bp; Villin-Cre = 567 bp; c. DAI index during DSS modeling; d. Survival at DSS modeling; e. Percentage change in body weight during DSS modeling; f. Colon length; g. Detection of mouse colonic permeability via the FITC–dextran assay; h. Mouse colon tissue. The values are expressed as the means ± SDs (n = 8). Superscript letters indicate significant differences at *P < 0.05, **P < 0.01, and ***P < 0.001.

CDH1+/- LAB group (P<0.001), whereas there was no significant difference in the intestinal permeability of the mice in the CDH1+/- LAB group compared with that of the CDH1+/- DSS group (Fig 1G).

The HE-stained tissue sections and histopathological scores of the mouse colon sections are shown in Fig 2A and 2B. The glands and pathological morphology of the colon were normal in the NC PBS group, and the epithelial cells were intact and neatly arranged, with the mucosa in an intact condition. However, the crypt structures of the CDH1+/- PBS group were loosely arranged, with slightly lower density. In the NC-DSS and CDH1+/- DSS groups, the colonic mucosa was incomplete, multiple focal shallow ulcers were visible, the number of goblet cells was reduced, most of the glands were disrupted, the surface epithelium was thoroughly damaged, and inflammation affected mainly the mucosa and submucosa. The inflammatory cells were composed mainly of neutrophils and were accompanied by a small number of lymphocytes and monocytes. The histological scores revealed that there was no significant difference between these two groups. The mice in the NC LAB group had neatly arranged and complete glands, the colon mucosa was not significantly eroded, and the goblet cells were relatively intact, with a reduced lesion range and severity and less inflammatory cell infiltration. Therefore, these mice had lower histological scores than did those in the NC DSS group (P<0.001). The CDH1+/- LAB group exhibited severe destruction of many crypts in their intestinal tissue, with a sharp decrease in goblet cells and increased inflammatory cell infiltration; moreover, the histological score was greater than that of the NC LAB group (P<0.001), whereas there was no difference compared with that of the CDH1+/- DSS group.

### 3.3. E-cadherin is involved in the anti-inflammatory effects of *L. gasseri* ATCC33323 in colitis

MPO activity strongly increases when the colon is severely damaged, and the penetration of immune cells in the colon tissue increases[28]. As shown in Fig 2C, the MPO activity was the highest in the NC DSS and CDH1+/- DSS groups, and there was no difference between the two groups. The level in the NC LAB group was significantly lower than that in the NC DSS group (P<0.001), and there was no statistically significant difference between the CDH1+/- DSS and CDH1+/- LAB groups. The MPO activity in the NC LAB group was significantly lower than that in the CDH1+/- LAB group (P<0.001).

SOD and MDA are both indices related to body oxidative damage. As shown in Fig 2D and 2E, there was no difference in SOD activity between the NC DSS and CDH1+/- DSS groups, and that in the NC LAB group was significantly greater than those in the NC DSS group (P<0.001) and the CDH1+/- LAB group (P<0.001). MDA activity was the highest in the NC DSS and CDH1+/- DSS groups, and there was no difference between the two groups. The level in the NC LAB group was lower than those in the NC DSS (P<0.001) and CDH1+/- LAB (P<0.001) groups. There was no significant difference in either SOD or MDA activity between the CDH1+/- DSS and CDH1+/- LAB groups.

The levels of the inflammatory factors IL-1β, IL-6, and TNFα in mouse intestinal tissue were studied to evaluate the extent of colitis-related inflammation. The results revealed that the inflammatory factors IL-1β, IL-6, and TNFα were expressed at the highest levels in the NC DSS and CDH1+/- DSS groups. The level of inflammatory factors in the NC LAB group was lower than those in the NC DSS (P<0.001, P<0.001, P<0.001) and CDH1+/- LAB groups (P<0.01, P<0.001, P<0.001); moreover, there was no statistically significant difference in the level of inflammatory factors between the CDH1+/- DSS and CDH1+/- LAB groups (Fig 2F).

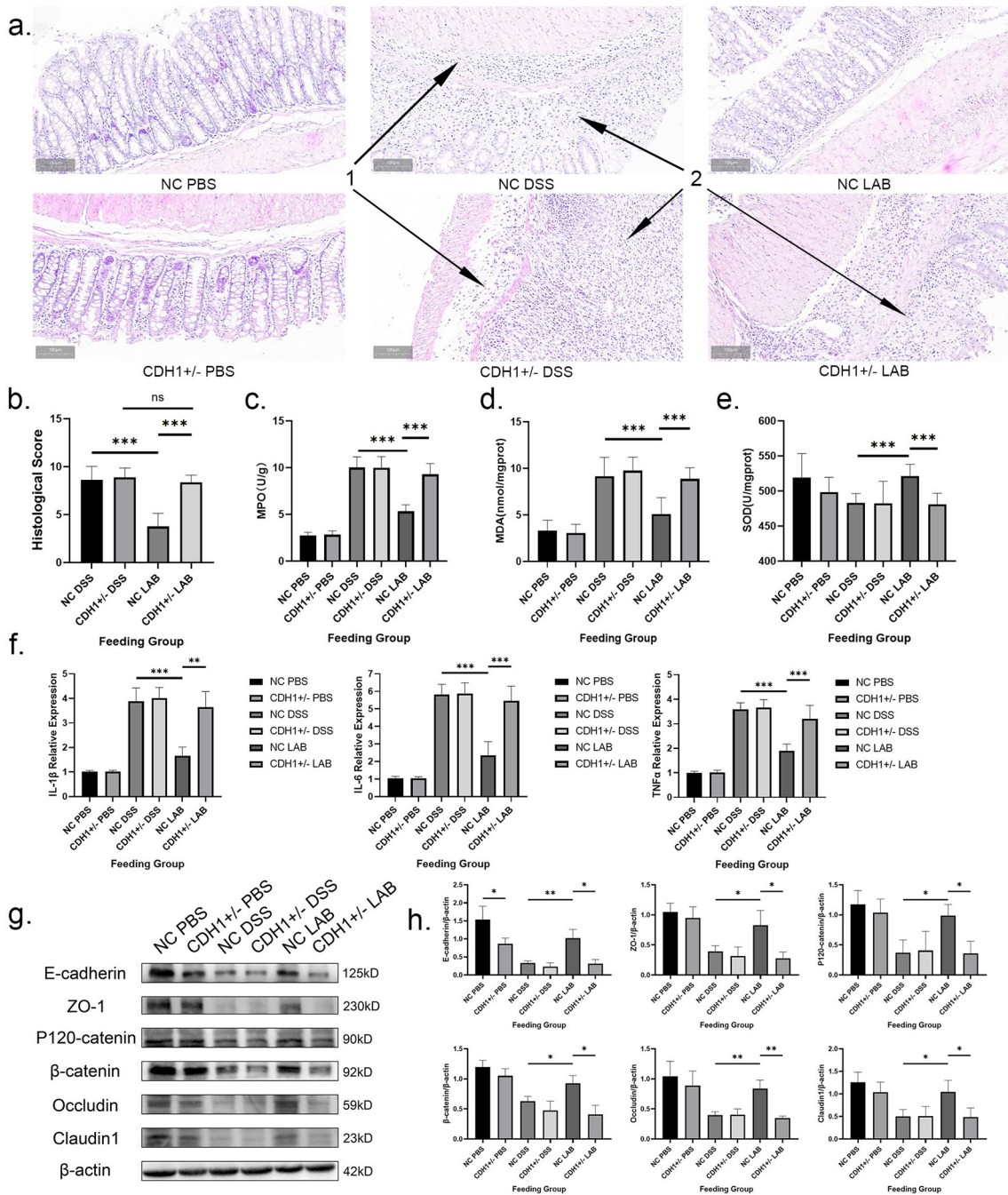

**Fig 2. Effects of *L. gasseri* ATCC33323 gavage on intestinal health, intestinal histopathology and inflammation during colitis induction.** a. Pathologic section of the mouse colon (H&E) at 200× magnification. 1. Inflammatory cell infiltration; 2. Disappearance of crypts; b. pathological scoring of colon severity; c. MPO activity; d. MDA activity; e. SOD activity; f. IL-1β, IL-6, and TNFα cytokines; g. Western blot validation of mucosal barrier protein expression; h. Statistical analysis of grayscale values of mucosal barrier protein western blot. The values are expressed as the means ± SDs (n = 8). Superscript letters indicate significant differences at *P < 0.05, **P < 0.01, and ***P < 0.001.

### 3.4. *L. gasseri* ATCC33323 affects junction protein expression through E-cadherin in colitis

The expression levels of barrier proteins in the colon mucosa of the mice were analyzed via western blot analysis. The results demonstrated that the expression levels of E-cadherin, ZO-1, claudin1, β-catenin, p120-catenin, and occludin were high in the NC PBS group. However, the expression of E-cadherin in the CDH1+/- PBS group was lower than that in the NC PBS group (P<0.05). The expression of junction proteins decreased after DSS treatment. *L. gasseri* ATCC33323 restored the expression of junction proteins in the NC LAB group (P<0.01, P<0.05, P<0.05, P<0.05, P<0.05, P<0.01), whereas the regulatory effect of *L. gasseri* ATCC33323 in the CDH1+/- LAB group was not significant (P<0.05, P<0.05, P<0.05, P<0.05, P<0.05, P<0.01) (Fig 2G and 2H).

Immunohistochemical staining was used to observe the localization and expression of E-cadherin, ZO-1, occludin, and claudin1 in colon tissue, and the CDH1+/- PBS group presented a decreased area of junction protein staining compared with the NC PBS group. These findings suggest that E-cadherin is fundamental for the maintenance of the intestinal epithelial structure and the aggregation of tight junctions. Owing to destruction of the intestinal epithelium, immune staining almost completely disappeared or included only a few cells at the top of the colon in the CDH1+/- DSS and NC DSS groups. The staining area and intensity of junction proteins in the NC LAB group were protected, which illustrated that the integrity of the intestinal epithelial barrier and the preservation and localization of junction proteins can be maintained by *L. gasseri* ATCC33323 (P<0.001, P<0.001, P<0.001, P<0.001). However, the staining of junction proteins was weak, and the increase in junction proteins was not significant in the CDH1+/- LAB group (Fig 3A, 3C, 3D, 3E and 3F).

The mucus barrier is formed after MUC2 is secreted from goblet cells in the intestine, and this barrier is also a protective layer of the intestinal mucosal epithelium[29]. The expression of MUC2 decreased due to the destruction of goblet cells in the CDH1+/- DSS and NC DSS groups. Compared with that in the NC DSS group, the damage to goblet cells in the NC LAB group was milder, and the expression of MUC2 was not affected (P<0.001). However, compared with that in the CDH1+/- DSS group, the staining intensity of MUC2 in the CDH1+/- LAB group was weaker because of the equally severe destruction of goblet cells (Fig 3B and 3F).

The regulation of intestinal mucosal barrier proteins can promote the rehabilitation of epithelial cells after injury[20], and both E-cadherin/β-catenin and E-cadherin/p120-catenin are important components of the intestinal mucosal barrier. Immunofluorescence staining also revealed that the NC LAB group had an intact epithelial mucosal barrier, and the presence of regular and strong green and red fluorescence indicated that the retention of these cells was greater than that in the NC DSS group (P<0.001, P<0.001). However, the green and red fluorescence signals corresponding to the combination of E-cadherin/β-catenin and E-cadherin/p120-catenin almost completely disappeared in the CDH1+/- LAB group and were not recovered by treatment with *L. gasseri* ATCC33323 (P<0.001, P<0.001) (Fig 4A, 4B, 4C and 4D).

### 3.5. Role of E-cadherin in the effects of *L. gasseri* ATCC33323 on gut microbes in mice with colitis

16S rDNA sequencing analysis of mouse stools was also conducted. In the α diversity analysis of microbes, the Chao1 index reflects the richness of species, whereas the Shannon and Simpson indices represent species diversity. Previous studies have confirmed a decrease in the diversity of the gut microbiota in IBD patients, and environmental homeostasis in the gut can be maintained with increasing microbiota diversity[30]. The present study revealed that there was no significant difference between the NC PBS and CDH1+/- PBS groups, whereas the

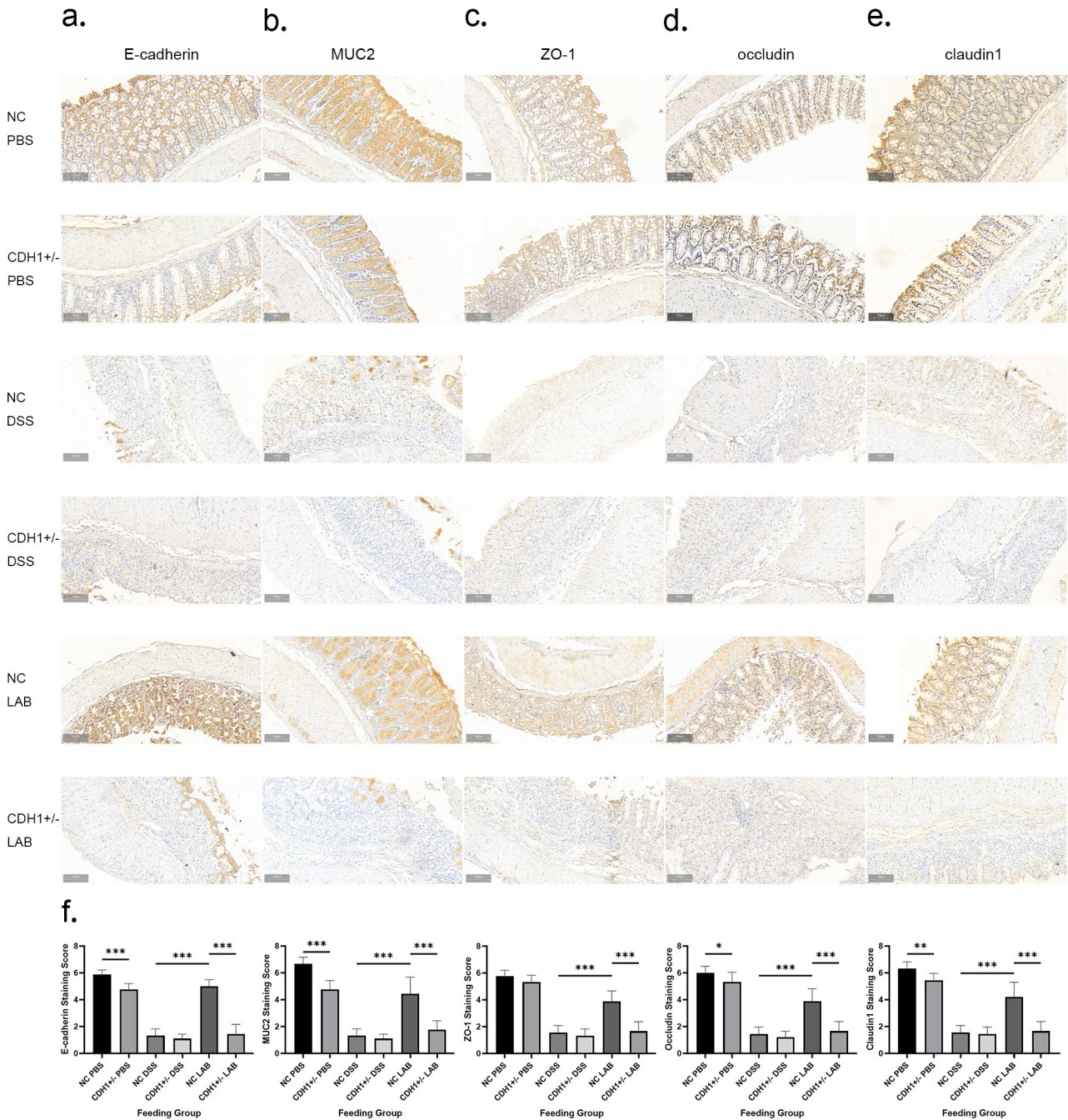

**Fig 3. Effects of *L. gasseri* ATCC33323 supplementation on the expression and localization of mucosal proteins in DSS-induced colitis.** Histopathological sections of the mouse colon were observed at 200× magnification. a. E-cadherin; b. MUC2; c. ZO-1; d. occludin; e. claudin1; f. Statistical analysis of immunohistochemical staining scores for mucosal barrier proteins.

Chao1, Shannon, and Simpson indices were significantly lower in both the NC DSS and CDH1+/- DSS groups. However, after treatment with *L. gasseri* ATCC33323, the Chao1, Shannon, and Simpson indices in the NC LAB group significantly recovered, whereas there was no improvement in the CDH1+/- LAB group (Fig 5A, 5B and 5C).

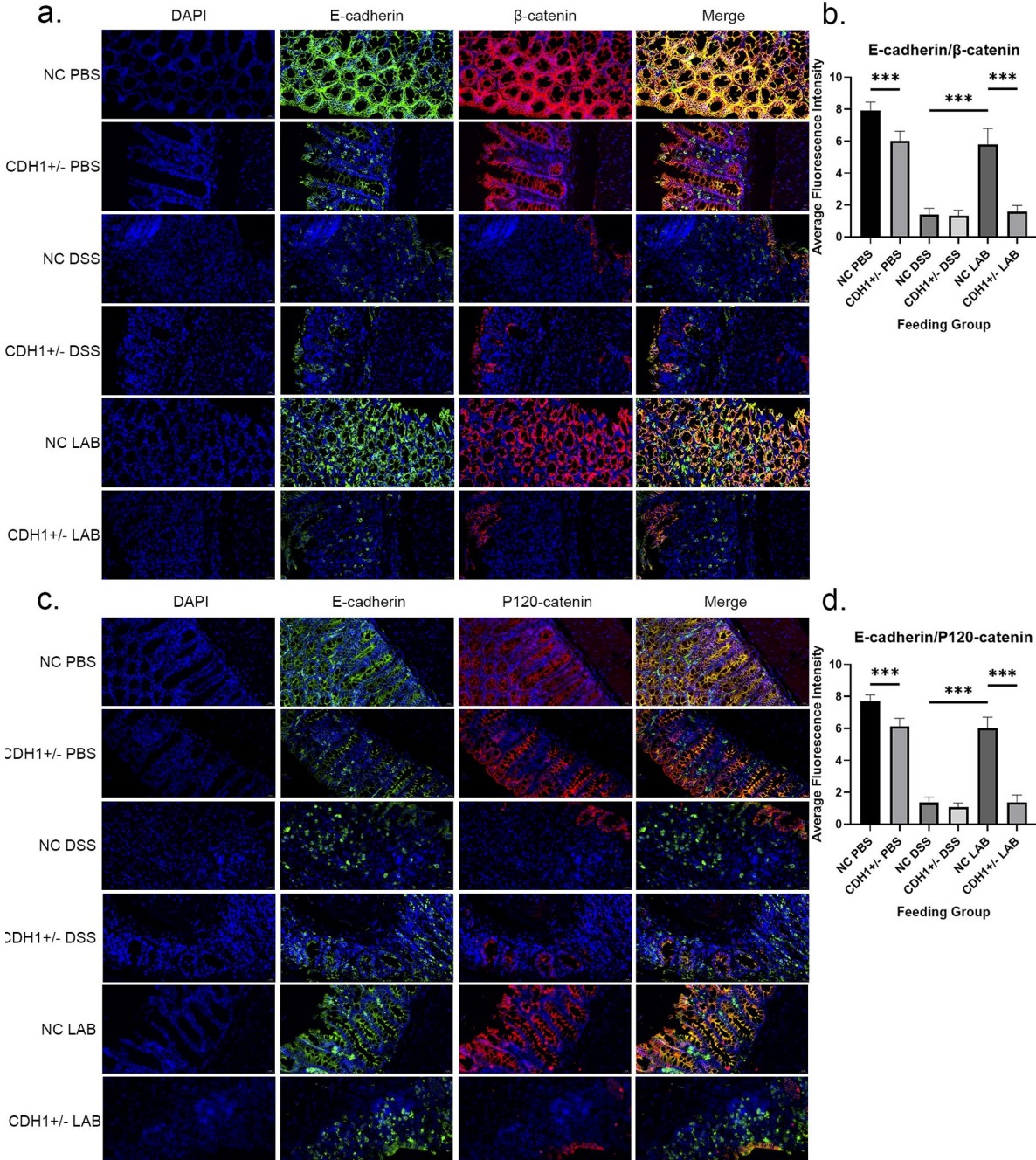

**Fig 4. Effects of *L. gasseri* ATCC33323 supplementation on the expression and localization of the E-cadherin/β-catenin complex and the E-cadherin/p120-catenin complex in DSS-induced colitis; magnification, 400×.** a. Fluorescence staining of the E-cadherin/β-catenin complex; b. Statistical analysis of the average fluorescence intensity of the E-cadherin/β-catenin complex; c. Fluorescence staining of the E-cadherin/p120-catenin complex; d. Statistical analysis of the average fluorescence intensity of the E-cadherin/p120-catenin complex.

PCA and PCoA were used to determine the species diversity among the different environmental communities via microbial β diversity analysis. Samples with high similarity in community structure were clustered together according to PCA and PCoA, and samples with significant differences in community structure were further separated. The results revealed

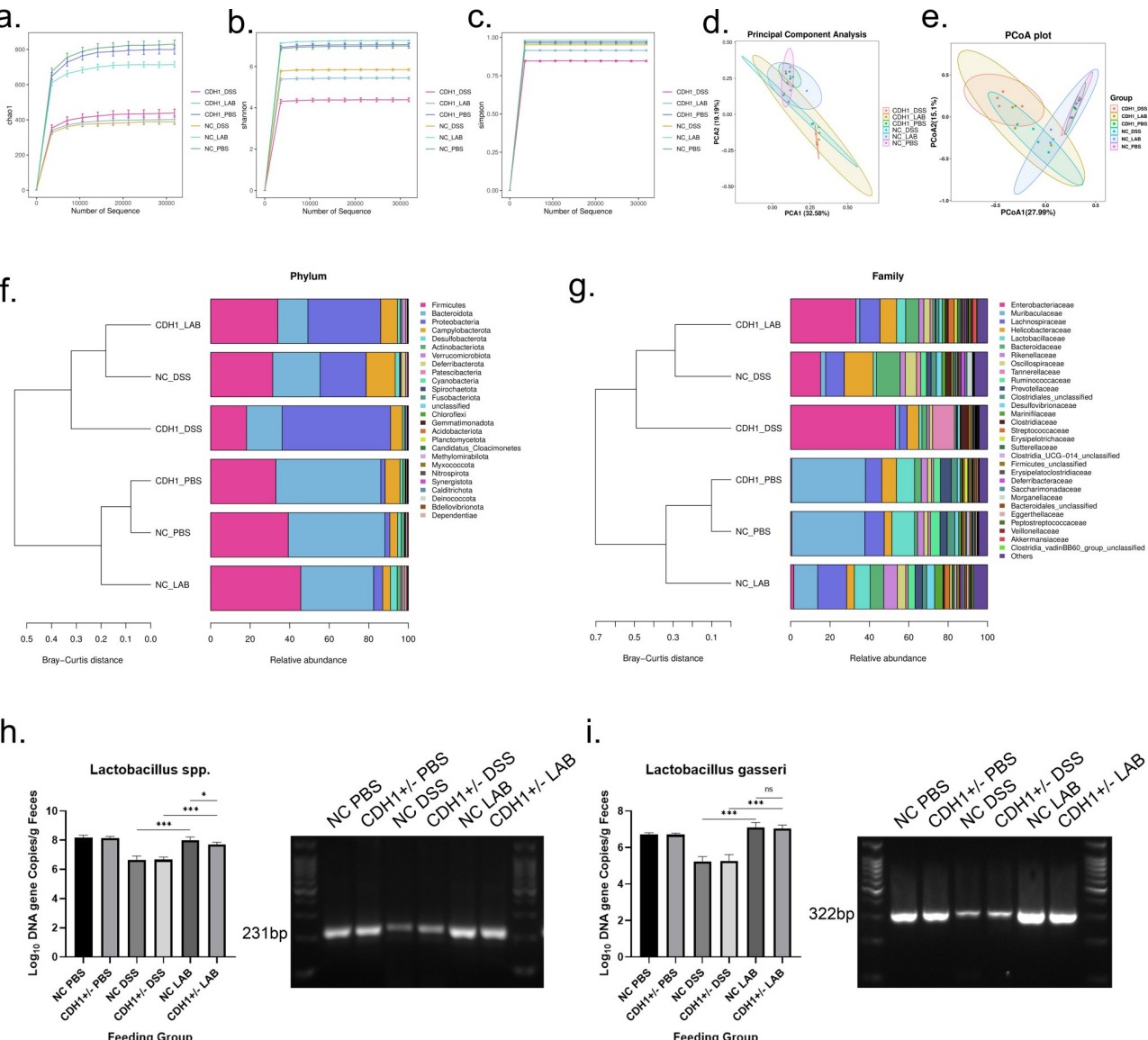

**Fig 5. 16S rDNA sequencing and qPCR were used to determine the effects of *L. gasseri* ATCC33323 on the intestinal microbiota of the mice, and the fecal bacteria of six groups of mice were analyzed.** a. Alpha diversity analysis of the mouse gut microbiota, Chao1 index; b. Alpha diversity analysis of the mouse gut microbiota, Shannon index; c. Alpha diversity analysis of the mouse gut microbiota, Simpson index; d. Beta diversity analysis of the mouse gut microbiota, PCA; e. Beta diversity analysis of the mouse gut microbiota, PCoA; f. composition of the gut microbiota at the phylum level among each group; g. composition of the gut microbiota at the family level among each group; h. determination of *Lactobacillus spp.* gene copy number levels in the DNA of each group of mouse fecal samples by qPCR and analysis of amplified samples by agarose gel electrophoresis; i. determination of *L. gasseri* ATCC33323 gene copy number levels in the DNA of each group of mouse fecal samples by qPCR and analysis of amplified samples by agarose gel electrophoresis.

that the regions in the NC PBS and CDH1+/- PBS groups highly overlapped and were concentrated, and the regions in the NC LAB group were significantly closer to those in the NC PBS and CDH1+/- PBS groups, whereas the regions in the CDH1+/- LAB group were similar to those in the NC DSS and CDH1+/- DSS groups (Fig 5D and 5E).

The relative abundance of a phylum was generally similar between the NC PBS group and the CDH1+/- PBS group, whereas the relative abundance of Firmicutes, Bacteroidota, Actinobacteria, and Fusobacteria decreased and the relative abundance of Proteobacteria and

Campylobacterota increased in the NC DSS and CDH1+/- DSS groups. After supplementation with *L. gasseri* ATCC33323, the distribution of bacterial communities in the NC LAB group improved, but there was no significant change in the CDH1+/- LAB group. Additionally, the distributions of the bacterial species in the NC PBS and CDH1+/- PBS groups were generally similar in terms of the relative abundance of a family, while the relative abundances of Enterobacteriaceae, Helicobacteraceae, and Tannerellaceae increased, whereas the relative abundances of Muribaculaceae, Lactobacillaceae and Rikenellaceae decreased in the NC DSS and CDH1+/- DSS groups. The bacterial distribution in the NC LAB group was restored; however, the relative abundance of Lactobacillaceae in the CDH1+/- LAB group was significantly greater than that in the CDH1+/- DSS group, possibly due to the effect of *L. gasseri* ATCC33323 gavage, whereas the relative abundance of other bacterial strains remained similar to that in the CDH1+/- DSS group (Fig 5F and 5G).

To investigate the effects of *L. gasseri* ATCC33323 on intestinal microorganisms, we quantified *Lactobacillus spp*. and *L. gasseri* ATCC33323 in the DNA of fecal samples from each group of mice to determine the efficacy of *Lactobacillus* transplantation. qPCR revealed that, at the *Lactobacillus spp*. level, after DSS treatment, the NC DSS and CDH1+/- DSS groups presented the lowest gene copy number of *Lactobacillus spp*., whereas when *L. gasseri* ATCC33323 was pretreated, the *Lactobacillus spp*. level in the NC LAB group was significantly greater than that in the NC DSS group (P<0.001), the *Lactobacillus spp*. level in the CDH1+/- LAB group was significantly greater than that in the CDH1+/- DSS group (P<0.001), and the *Lactobacillus spp*. level was also greater in the NC LAB group than in the CDH1+/- LAB group (P<0.05) (Fig 5H). After determining the level of *L. gasseri* ATCC33323, we found that the gene copy number of *L. gasseri* ATCC33323 was significantly greater in the NC LAB group than in the NC DSS group (P<0.001) and was significantly greater in the CDH1+/- LAB group than in the CDH1+/- DSS group (P<0.001); the NC LAB group *L. gasseri* ATCC33323 levels were not significantly different from those in the CDH1+/- LAB group (Fig 5I).

### 3.6. Comparative analysis of RNA-seq data from mouse colon tissue

RNA-seq was performed on the NC DSS, CDH1+/- DSS, CDH1+/- LAB, and NC LAB groups. Compared with the NC LAB group, the NC DSS group presented 980 significantly upregulated genes and 915 significantly downregulated genes (Fig 6A). Compared with the CDH1+/- LAB group, the NC LAB group presented 1003 significantly upregulated genes and 545 significantly downregulated genes (Fig 6B). The gene expression levels of specific samples were visualized through heatmaps (Fig 6C).

The GO analysis revealed that the DEGs in the NC DSS and NC LAB groups were involved in biological processes such as the cell cycle, cell division, the cellular response to DNA damage stimulus, and DNA repair; cellular components such as the nucleus, cytoplasm, nucleoplasm, and membrane; and molecular functions such as protein binding, nucleotide binding, ATP binding, and metal ion binding (Fig 6D). Additionally, the DEGs in the NC LAB and CDH1 +/- LAB groups were shown to be involved in biological processes such as the cell cycle, DNA replication, multicellular organism development, and other biological processes; cellular components such as the membrane, integral component of the membrane, nucleus, and cytoplasm; and molecular functions such as protein binding, metal ion binding, nucleotide binding, and DNA binding (Fig 6E).

The KEGG analysis results revealed that the DEGs in the NC DSS and NC LAB groups were enriched mainly in the following disease-related pathways: the cell cycle, oocyte meiosis, DNA replication, and base excision repair (Fig 6F); however, the disease-related pathways associated with the DEGs in the NC LAB and CDH1+/- LAB groups included DNA

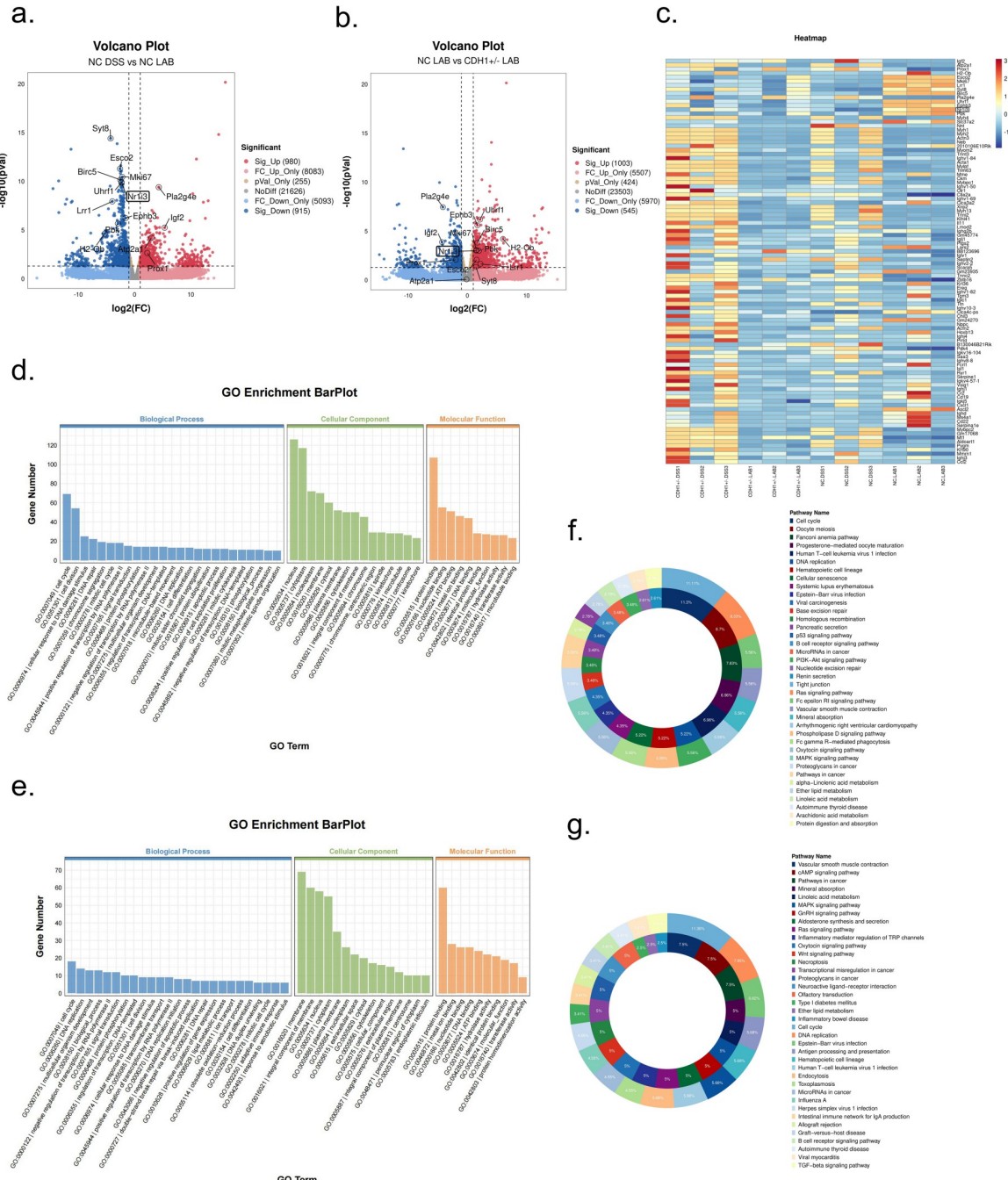

**Fig 6. RNA-seq analysis of colon tissue.** a. Volcano plots were used to analyze differential genes between the NC DSS and NC LAB groups; b. Volcano plots were used to analyze differential genes between the NC LAB and CDH1+/- LAB groups; c. Heatmaps were used to analyze the differential genes between the NC DSS, NC LAB, CDH1+/- DSS, and CDH1+/- LAB groups; d. GO enrichment analysis between the NC DSS and NC LAB groups; e. GO enrichment analysis between the NC LAB and CDH1+/- LAB groups; f. KEGG enrichment analysis between the NC DSS and NC LAB groups; g. KEGG enrichment analysis between the NC LAB and CDH1+/- LAB groups.

replication, the cell cycle, the intestinal immune network for IgA production and inflammatory bowel disease (Fig 6G).

### 3.7. E-cadherin expression in cells was promoted by *L. gasseri* ATCC33323 through NR1I3

On the basis of the differential genes in the RNA-seq results, we selected *ESCO2*, *MKI67*, *SYT8*, *BIRC5*, *PLA2G4E*, *UHRF1*, and *NR1I3* among the top 25 genes in terms of the q-values, as well as other meaningful differential genes that have been reported in the literature to be related to inflammation or the intestines, *IGF2*, *ATP2A1*, *PROX1*, *H2-OB*, *LRR1*, *EPHB3*, and *PBK[31–37]*. Caco2 and HCT116 colon cancer cells were cocultured with *L. gasseri* ATCC33323 for 3, 6, or 12 hours before qPCR validation was conducted. The expression of the *CDH1* gene in colon cancer cells increased with prolonged intervention with *L. gasseri* ATCC33323. Similarly, according to the qPCR results, NR1I3 expression also increased continuously in response to *L. gasseri* ATCC33323, which was consistent with the RNA-seq results (Fig 7A).

After selecting NR1I3, we performed another validation via western blot experiments in the Caco2 and HCT116 cell lines. The expression of NR1I3 gradually increased with increasing coculture time under stimulation with *L. gasseri* ATCC33323. Additionally, E-cadherin was upregulated with increasing coculture time, which was consistent with the trend observed for NR1I3 (Fig 7B).

The expression of endogenous NR1I3 in cells was downregulated by siRNA to further confirm the importance of NR1I3 in regulating E-cadherin function. The HCT116 and Caco2 cell lines were transfected with negative control siRNA oligo as the control group, and the HCT116 and Caco2 cell lines were transfected with two siRNA oligos, NR1I3-homo427 and NR1I3-homo981, as the experimental group. The results demonstrated that the expression of E-cadherin also decreased when NR1I3 was knocked out (Fig 7C). Reduced intracellular expression of NR1I3 can passivate the expression of E-cadherin. *L. gasseri* ATCC33323 was subsequently cultured in these two cell lines for 3, 6, and 12 hours, and the increase in E-cadherin expression with increasing *L. gasseri* ATCC33323 coculture time was significantly attenuated after NR1I3 knockdown (Fig 7D).

### 3.8. Regulation of the expression of E-cadherin through the binding of the *CDH1* promoter and NR1I3

The RNA-seq and bioinformatics analyses indicated that NR1I3 may regulate *CDH1* expression, and thus ChIP and dual-luciferase reporter genes were used to verify this possibility. Initially, the binding site sequences of the *CDH1* promoter that may be regulated by NR1I3 were selected through JASPAR (https://jaspar.elixir.no/analysis), which ranged from -939 to -931 bp, and primers were designed on the basis of these fragments. ChIP was used to verify the interaction between the *CDH1* gene promoter and NR1I3 in both the HCT116 and Caco2 cell lines. Compared with those in the negative control samples, the amplified fragments in the samples treated with the anti-NR1I3 antibody were significantly enriched, and DNA aggregation was increased relative to that in the immunoglobulin precipitation samples ($P<0.001$, $P<0.05$) (Fig 7F), which also confirmed the interaction between the *CDH1* promoter and NR1I3. The sizes of the DNA fragments were subsequently analyzed via agarose gel electrophoresis (Fig 7E).

To evaluate whether NR1I3 can regulate the activity of the *CDH1* promoter, we conducted an in-depth study on the transcriptional ability of NR1I3 via dual-luciferase reporter gene experiments. The *CDH1*-promoter-wt plasmid and the *CDH1*-promoter-mut plasmid were cotransfected with the NR1I3 overexpression plasmid. The transcriptional activation ability of the construct was detected, and the results revealed that the luciferase activity of the *CDH1* wild-type vector was significantly greater than that of the *CDH1* promoter mutant vector and

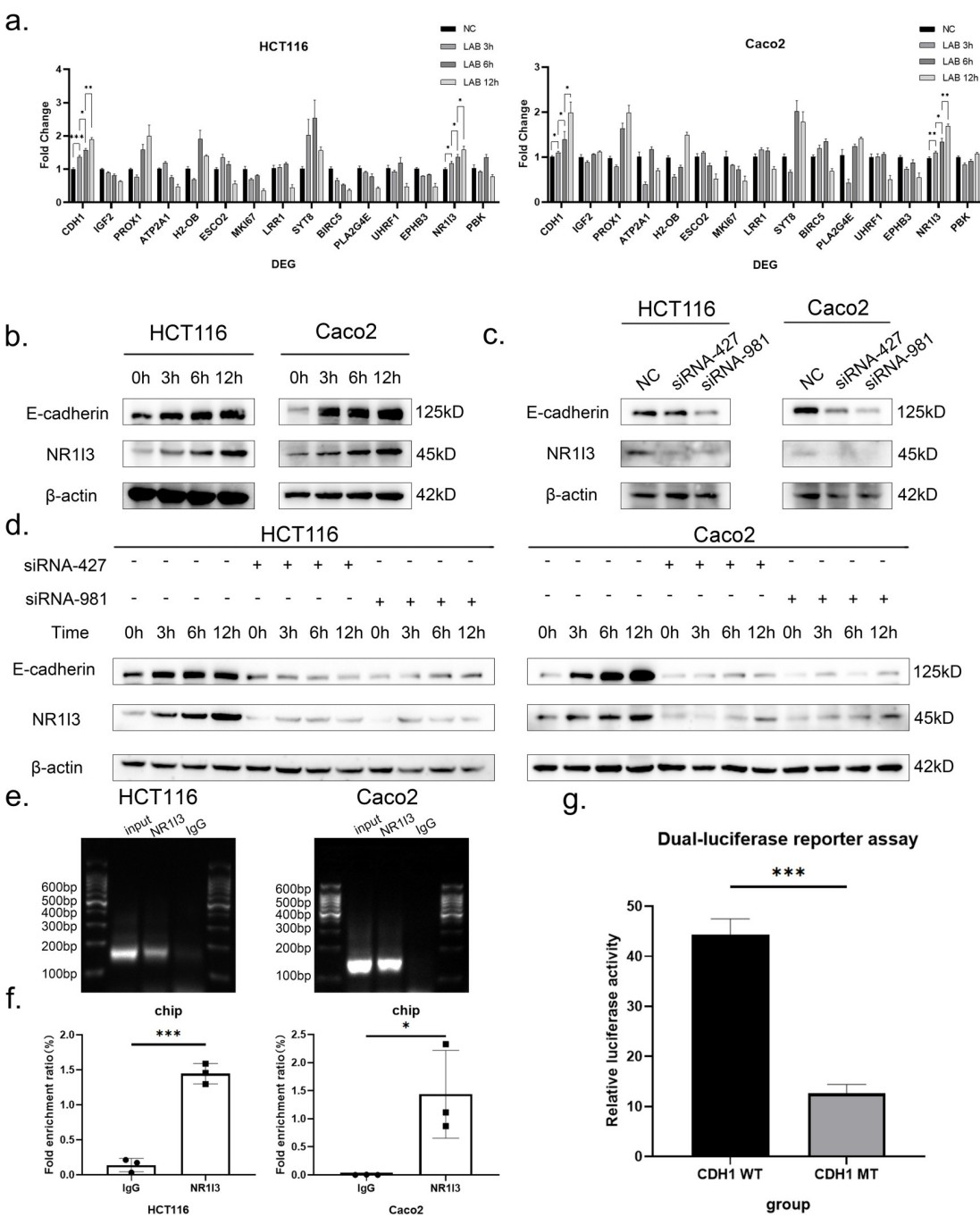

**Fig 7.** *In vitro* **study of** *L. gasseri* **ATCC33323 with colon cancer cell lines.** a. qPCR assay to analyze the differential gene expression of *L. gasseri* ATCC33323 after 3/6/12 h of coculture with HCT116 and Caco2 colon cancer cells; b. Western blot analysis of NR1I3 and E-cadherin expression in *L. gasseri* ATCC33323 after 3/6/12 h of coculture with HCT116 and Caco2 colon cancer cells; c. Validation of the knockdown efficiency of NR1I3 in HCT116 and Caco2 colon cancer cells; d. NR1I3 and E-cadherin expression after intracellular NR1I3 knockdown in coculture with *L. gasseri* ATCC33323 for 3/6/12 h; e.,f. Results of ChIP experiments showing that NR1I3 can act as a transcription factor for CDH1; g. Luciferase reporter gene assay to verify the activity of NR1I3 as a CDH1 transcription factor. The values are expressed as the means ± SDs (n = 3). Superscript letters indicate significant differences at $^*P < 0.05$, $^{**}P < 0.01$, and $^{***}P < 0.001$.

that the expression was nearly fourfold greater (P<0.001) (Fig 7G). The -939~-931 bp sequence in the *CDH1* promoter region contains a highly conserved regulatory site in NR1I3. Mutation of this cis-element significantly reduced the activity of the *CDH1* promoter, which was induced by NR1I3, and the downstream reporter gene of the *CDH1* promoter was activated only when NR1I3 and the *CDH1* wild type were cotransfected.

### 3.9. *L. gasseri* ATCC33323 affects the anti-inflammatory effects of LPS-stimulated cells through NR1I3

Lipopolysaccharide (LPS) is an important component of the outer membrane of *Escherichia coli* and can cause damage to the intestinal epithelial barrier and induce chronic intestinal inflammation[19]. Stimulation with LPS induces an inflammatory response and decreases E-cadherin expression in HCT116 and Caco2 cells. However, pretreatment of the cells with $1 \times 10^8$ cfu/mL *L. gasseri* ATCC33323 resulted in a significant increase in E-cadherin expression (P<0.01). When we treated NR1I3 in the cell line with the two interfering sequences separately, we found that the expression of E-cadherin was significantly attenuated (P<0.01) in both groups compared with that in the cells pretreated with *L. gasseri* ATCC33323 (Fig 8A and 8B).

The expression of inflammatory factors was investigated to assess the inflammatory effects of *L. gasseri* ATCC33323 on LPS-stimulated cells. In the HCT116 and Caco2 cell lines, the gene expression of the proinflammatory factors IL-1β, IL-6, and TNFα was significantly lower after pretreatment with *L. gasseri* ATCC33323 than in the positive control LPS group (P<0.001). When NR1I3 was knocked down in the cells by siRNA, the expression of IL-1β, IL-6 and TNFα returned to extremely high levels (P<0.01) (Fig 8C and 8D). Thus, *L. gasseri* ATCC33323 exerts anti-inflammatory effects through NR1I3.

## 4. Discussion

Inflammatory bowel disease (IBD) is an immune system disease characterized by rectitis and colitis and comprises ulcerative colitis (UC) and Crohn's disease (CD) with or without systemic symptoms such as diarrhea, abdominal pain, and bloody stools[38]. Probiotics are defined as living microorganisms that benefit the health of the host when sufficient amounts are given according to the definition of the World Food and Agriculture Organization (FAO)/World Health Organization (WHO)[39]. *Lactobacillus* species are closely related to the health of the intestine and have begun to be widely used as probiotics to improve intestinal health.

*Lactobacillus gasseri* is a common symbiotic bacterium that is cultivated from microorganisms of the human body and is also one of the main microorganisms cultivated in the gastrointestinal tract of newborns at the initial stage[40]. *Lactobacillus gasseri* strains are resistant to bile and exhibit antibacterial ability, degradation of oxalic acid, immune regulation, and adhesion to Caco2 intestinal epithelial cells[41–43]. We chose *Lactobacillus gasseri* to test its impact on colitis given its enormous potential.

Intercellular junctions are key factors for tissue integrity, and the apical junction complex of the intestinal epithelium consists of tight junctions, adhesive junctions, and desmosomes [44]. TJs have a continuous and circular structure and can also form an osmotic barrier at the top of cell gaps. AJs are near TJs and play important roles in cell recognition and the regulation of intercellular connections. Desmosomes are located below AJs and are punctate intercellular junctions[45]. The three key components of TJs are the claudin family, ZO-1, and occludin. ZO-1 is connected to cytoskeletal actin after binding to the transmembrane protein occludin and claudins, and cytoskeletal actin contraction plays a crucial role in the regulation of cell bypass permeability[46, 47]. The complex formed by cadherins and catenins in AJs can promote the aggregation of TJs[48]. The main component involved in adhesive connections is E-

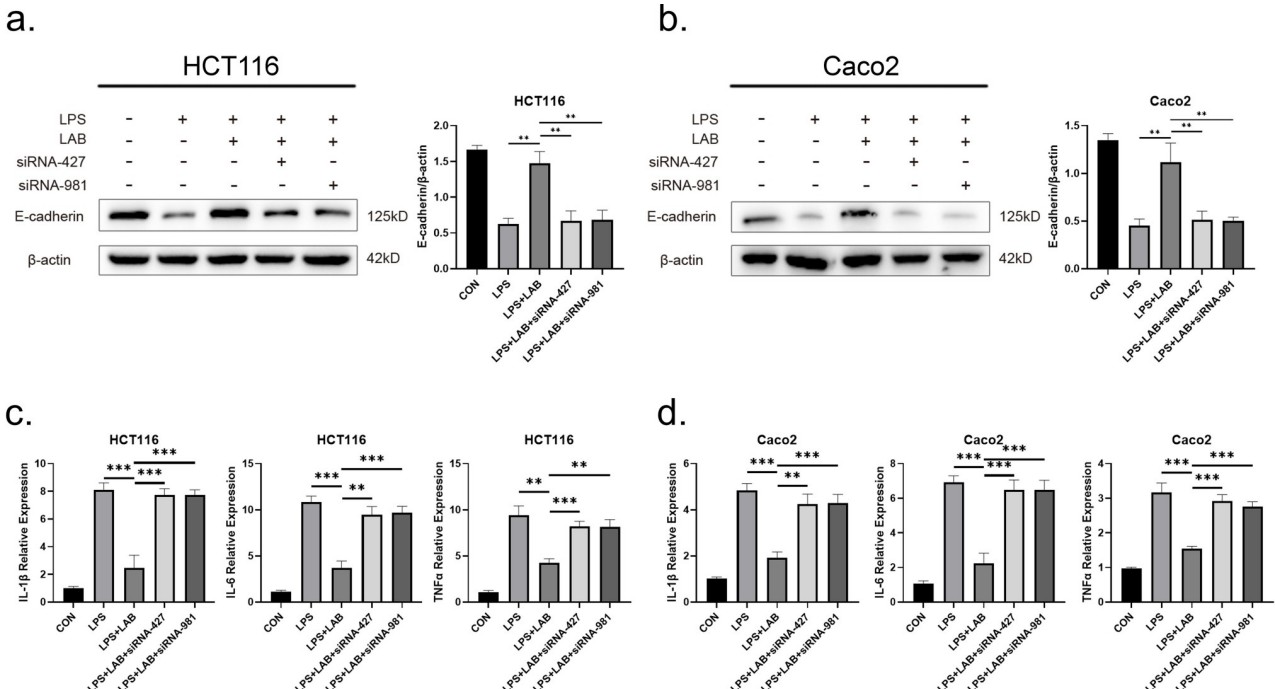

**Fig 8. Effects of *L. gasseri* ATCC33323 supplementation on LPS-stimulated cellular inflammatory responses.** a. E-cadherin expression in HCT116 cell lines with or without NR1I3 knockdown by *L. gasseri* ATCC33323 pretreatment; b. E-cadherin expression in Caco2 cell lines with or without NR1I3 knockdown by *L. gasseri* ATCC33323 pretreatment; c. IL-1β, IL-6 and TNFα expression in HCT116 cell lines with or without NR1I3 knockdown by *L. gasseri* ATCC33323 pretreatment; d. IL-1β, IL-6 and TNFα expression in Caco2 cell lines with or without NR1I3 knockdown by *L. gasseri* ATCC33323 pretreatment. The values are expressed as the means ± SDs (n = 3). Superscript letters indicate significant differences at $^*P < 0.05$, $^{**}P < 0.01$, and $^{***}P < 0.001$.

cadherin, which participates in hydrophilic cell–cell interactions, binds to catenin within the cell (β-catenin, plakoglobin, and p120-catenin), and subsequently connects transmembrane E-cadherin to the actin skeleton through α-catenin[49].

Loss of E-cadherin function in the intestine is often related to pathological processes, and studies have shown that there is a decrease in E-cadherin expression in IBD patients[50]. E-cadherin has multiple functions in the process of tissue morphogenesis and is essential for embryonic development[51]. Studies have shown that E-cadherin-mediated adhesion is a necessary prerequisite for the formation of cell junctions. The complete absence of E-cadherin results in the absence of or significant reduction in α-catenin, β-catenin, and p120-catenin [52]. The permeability of tight junctions is altered when E-cadherin is mutated in mouse skin, and the tight junction proteins ZO-1, claudin1, and occludin may either be missing or no longer function normally[53]. Schneider et al. proposed removing E-cadherin from the intestinal epithelium of adult mice by using *Villin-Cre*, which was induced by tamoxifen, and hemorrhagic diarrhea was observed prior to euthanasia of animals lacking E-cadherin[25]. Benjamin J reported that mice with E-cadherin conditional gene knockout (CKO) in the intestinal epithelium were unable to survive, which also caused the integrity of tight junctions to suffer injuries and reduced the expression of claudin1. Activated β-catenin protein, which was mutated by E-cadherin and significantly lower than that in the control tissue, was detected in intestinal extracts. A large gap was observed between the epithelial cells of the intestinal tissue, which was CKO by E-cadherin, via electron microscopy, whereas the control epithelial cells were tightly arranged[54].

Therefore, the purpose of this study was to explore whether *Lactobacillus gasseri* acts on E-cadherin to ameliorate colitis in mice and to discover the exact mechanism involved. We constructed E-cadherin semiknockout mice with the *Villin-Cre;Cdh1^{wt/flox}* genotype because complete E-cadherin conditional gene knockout in the intestine can cause mouse death. DSS was used to establish a mouse model of colitis[55]. We evaluated the effects of *L. gasseri* ATCC33323 on colitis in mice by evaluating changes in physiological damage, pathological damage, inflammation severity, mucosal barrier proteins, and the microbiota.

We first found that both *WT mice* and *Cdh1* mutant mice develop very heavy colitis in the later stages of DSS-induced colitis. Moreover, colitis symptoms and colon shortening can be significantly alleviated in *WT* mice following *L. gasseri* ATCC33323 intervention, whereas *L. gasseri* ATCC33323 does not have a good therapeutic effect on *Cdh1* mutant mice. Histopathologically, the semiknockout of E-cadherin in the intestine revealed a sparser arrangement of intestinal epithelial cells. The infiltration of inflammatory cells, such as neutrophils and macrophages, is a hallmark of IBD pathology. Tissue destruction and neutrophil infiltration were significantly reduced, and the integrity of the intestinal mucosa was preserved in DSS-induced *WT* mice treated with *Lactobacillus gasseri* ATCC33323. In contrast, the protective effect on *Cdh1* mutant mice was low.

Cytokines are proteins that can regulate immunity and inflammation. TNFα, IL-1β, and IL-6 participate in the proinflammatory regulation of IBD progression[27]. MPO is a biomarker of acute intestinal inflammation caused by the abundant expression of neutrophils, while MDA and SOD levels reflect the body's oxidative damage and antioxidant capacity[20]. After DSS treatment, the expression of proinflammatory factors and the severity of oxidative damage were not significantly different between the E-cadherin knockout and WT mice. Whereas *Lactobacillus gasseri* ATCC33323 effectively improved colitis in E-cadherin WT mice by enhancing their antioxidant damage capacity and anti-inflammatory ability, it did not significantly improve colitis in E-cadherin knockout mice.

The relationship between *Lactobacillus gasseri* ATCC33323 and intestinal barrier function was explored. E-cadherin serves as the basis of junction protein expression, and when E-cadherin was knocked out in the mouse intestine, the expression of the junction proteins ZO-1, MUC2, claudin1, occludin, β-catenin, and p120-catenin was somewhat attenuated as the arrangement of the intestinal epithelium dispersed. In contrast, after DSS induction, the intestinal structure was severely disrupted, and with damage to the mucosal epithelium, junction proteins presented diminished expression and loss of localization. In WT colitis model mice, *L. gasseri* ATCC33323 preserved the presence and maintains the expression of junction proteins. Because a correlation has also been found between junction protein expression and intestinal permeability[56], intestinal wall permeability was also protected in mice, which provides new insights into the pathways involved in strengthening the mucosal barrier. In E-cadherin-knockout colitis model mice, junction protein expression was not restored by *L. gasseri* ATCC33323 due to E-cadherin deletion, and this change was accompanied by increased intestinal permeability.

16SrDNA revealed that *L. gasseri* ATCC33323 can improve the abundance of the gut microbiota and microbial diversity in WT colitis model mice. The ability of *L. gasseri* ATCC33323 to correct microbial flora was weakened when E-cadherin was knocked out. By observing the composition and expression of the gut microbiota, we found that *L. gasseri* ATCC33323 can increase the abundance of several harmful bacteria and ameliorate the decrease in the abundance of several beneficial bacteria in WT colitis model mice at the phylum and family levels. However, we found that the abundance of Lactobacillaceae in E-cadherin semiknockout colitis model mice treated with *L. gasseri* ATCC33323 was still greater than that in control mice at the family level. We speculate that this phenomenon was due to the effect of *L. gasseri* ATCC33323

gavage, while the relative abundances of the other strains were not significantly different from those in mice treated with DSS. Next, we analyzed the DNA in mouse fecal samples for absolute quantification of gene copy number via a qPCR assay to demonstrate the efficacy of microbial transplantation and found that pretreatment with *L. gasseri* ATCC33323 resulted in an increase in the levels of *L. gasseri* ATCC33323 and *Lactobacillus spp*. in the intestinal tracts of the mice. Although E-cadherin knockout colitis mice still had higher levels of *Lactobacillus spp*. after gavage than DSS-treated mice did, we attributed this to *L. gasseri* ATCC33323 treatment, which did not ameliorate intestinal dysbiosis, and the *Lactobacillus spp*. level was still lower than that in WT mice.

Although *L. gasseri* ATCC33323 can reduce inflammatory damage in the mouse intestine and protect the intestinal mucosal barrier, the specific mechanism involved is still unclear. We performed RNA-seq on the colon tissue of mice, selected several genes for *in vitro* validation via RNA-seq and analyzed whether the selected genes were consistent with the *in vivo* research results. The RNA-seq results revealed that the expression of some proto-oncogenes increased and that of some tumor suppressor genes decreased in the NC LAB group, whereas the opposite results were obtained *in vitro*; moreover, *in vitro* studies have shown that *Lactobacillus gasseri* can inhibit the proliferation of colon cancer cells[57]. Since these genes may be expressed in the vast majority of cells, their expression in other structural cells of the intestine may affect the RNA-seq results. Moreover, most of these oncogenes are related to the cell cycle; therefore, it is necessary to discuss both the long-term and short-term outcomes associated with these genes. Intestinal epithelial cells gradually undergo apoptosis and significantly decrease in number during the process of intestinal barrier injury, which results in the expression of several genes involved in cell proliferation, such as MKI67 and PBK, or cell apoptosis resistance, such as BIRC5. Therefore, short-term changes in the expression of these oncogenes and tumor suppressor genes may promote the proliferation of epithelial cells[58].

We selected NR1I3, which is responsible for encoding the conservative androstane receptor (CAR), according to the RNA-seq, bioinformatics analysis, and PCR screening results. CAR, which is encoded by NR1I3, is an important nuclear receptor (NR) that is highly expressed in the liver and intestines. CAR and PXR are members of the same NR1I nuclear receptor subfamily, and the relationships between these receptors and the immune system, environmental factors, and the gut microbiota are closely related[59]. Hudson et al. recently reported that, in the intestinal mucosal biopsy tissues of IBD patients with active mild/moderate inflammation, the expression of the NR1I3 gene was significantly lower than that in healthy control samples from patients with noninflammatory bowel disease. The expression of NR1I3 in colon tissue was significantly reduced in the mice that were induced by DSS. When mice were gavaged with the CAR agonist TCPOBOP, the colon MPO levels were significantly reduced, mucosal healing was enhanced, inflammation resolution was accelerated, and histological scores were reduced[60]. NR1I3 can be regulated by metabolites of the microbiota and thus affect intestinal inflammation[61], which indicates that their functions may also provide additional space for host microbial interactions. In addition, CAR can be pharmacologically activated by TCPOBOP and subsequently affect the composition of the intestinal microbiota[62]. Therefore, we believe that the relationship between NR1I3 and the microbiota in the intestine is mutually influential and interactive.

We speculate that *L. gasseri* ATCC33323 can regulate the NR1I3 pathway to affect the expression of E-cadherin and subsequently improve intestinal barrier function. Overall, although both the NC LAB group and the CDH1+/- LAB group were treated via gavage, the expression of the NR1I3 gene in the CDH1+/- LAB group did not significantly increase; thus, we speculate that the gut microbiota had not fully recovered and that more harmful bacteria

existed. The metabolic products of other harmful bacteria may reduce the expression of NR1I3, but the specific mechanism requires further research.

We designed *in vitro* experiments to test if *L. gasseri* ATCC33323 can regulate E-cadherin through NR1I3 and validated these findings in the colon cancer cell lines Caco2 and HCT116. First, we cocultured two cell lines with *L. gasseri* ATCC33323 and found that the expression of NR1I3 and E-cadherin increased over time with increasing coculture time. To verify the role of NR1I3, we knocked out NR1I3 in cells via siRNA and found that the expression of E-cadherin also decreased when endogenous NR1I3 was lost in the cell lines. *L. gasseri* ATCC33323 was subsequently cocultured with Caco2 and HCT116 cells in which NR1I3 was knocked out, and the increase in E-cadherin expression was significantly weakened. To investigate the specific mechanism by which NR1I3 regulates E-cadherin expression, we used ChIP experiments to verify whether NR1I3 can bind to *CDH1* by regulating its transcription, and the transcriptional activity of NR1I3 was proven through dual-luciferase reporter gene experiments. Finally, we confirmed that NR1I3 can bind to the -939~-931 bp sequence of the *CDH1* promoter and affect the expression of E-cadherin.

It is well known that LPS stimulates proinflammatory cytokines, mainly IL-1β, IL-6 and TNF-α. The production of proinflammatory cytokines via intestinal epithelial cells can lead to impaired intestinal barrier function[63]. Therefore, LPS is often used to model inflammation-related conditions. Our results revealed that *L. gasseri* ATCC33323 effectively inhibited LPS-stimulated inflammatory responses by decreasing the levels of IL-1β, IL-6, and TNFα, whereas the anti-inflammatory effect of *L. gasseri* ATCC33323 was significantly attenuated when we interfered with NR1I3 in the cell lines.

## 5. Conclusion

This study revealed that *L. gasseri* ATCC33323 can ameliorate physiological damage in mice with colitis, reduce the extent of colitis, decrease inflammatory cell infiltration, and decrease the levels of the inflammatory factors IL-1β, IL-6, and TNFα. In addition, *L. gasseri* ATCC33323 can also protect the integrity of the intestinal epithelial barrier and increase the expression of junction proteins, such as E-cadherin, MUC2, ZO-1, claudin1, β-catenin, p120-catenin, and occludin. *L. gasseri* ATCC33323 can retain the localization of the E-cadherin/β-catenin complex and the E-cadherin/p120-catenin complex, reduce the permeability of the intestinal wall, and regulate the activities of MPO, MDA, and SOD. Moreover, *L. gasseri* ATCC33323 can ameliorate intestinal microbiota disorders and affect intestinal health. When we semiknocked out E-cadherin from the intestines of the mice, the regulatory ability of *L. gasseri* ATCC33323 was significantly reduced. The importance of E-cadherin in the pathogenesis of IBD is clear. We confirmed that *L. gasseri* ATCC33323 can regulate the expression of E-cadherin through the regulation of CDH1 transcription, which is affected by NR1I3. *In vitro* experiments demonstrated that *L. gasseri* ATCC33323 can ameliorate the inflammatory expression of cells, possibly through NR1I3. This provides a new direction for *Lactobacillus* strains to improve colitis treatment efficacy and provides us with a deeper understanding of the use of *Lactobacillus* strains in treating IBD. However, further research is needed on the specific mechanism through which *L. gasseri* ATCC33323 affects NR1I3, and additional experiments may be needed to validate the current findings.

## Supporting information

**S1 Table. Sequences of the primers used for real-time reverse transcription PCR.**
(DOCX)

**S1 Fig. Fig 7 Western blot gray value analysis.** a. Analysis of E-cadherin and NR1I3 grayscale values in the HCT116 cell line in Fig 7B; b. Analysis of E-cadherin and NR1I3 grayscale values in the Caco2 cell line in Fig 7B; c. Analysis of E-cadherin and NR1I3 grayscale values in the HCT116 cell line in Fig 7C; d. Analysis of E-cadherin and NR1I3 grayscale values in the Caco2 cell line in Fig 7C; e. Analysis of E-cadherin and NR1I3 grayscale values in the HCT116 cell line in Fig 7D; f. Analysis of E-cadherin and NR1I3 grayscale values in the Caco2 cell line in Fig 7D. The values are expressed as the means ± SDs (n = 3). Superscript letters indicate significant differences at $^*P < 0.05$, $^{**}P < 0.01$, and $^{***}P < 0.001$.
(TIF)

**S2 Fig. Raw images of western blot for E-cadherin, ZO-1, p120-catenin, β-catenin, occludin, claudin1, and β-actin.**
(TIF)

**S3 Fig. Raw images of western blot for E-cadherin, NR1I3, and β-actin.**
(TIF)

# Author Contributions

**Conceptualization:** Guanru Qian, Hui Zang, Jingtong Tang, Hao Zhang.

**Data curation:** Guanru Qian.

**Formal analysis:** Guanru Qian.

**Funding acquisition:** Jianping Zhou.

**Investigation:** Guanru Qian, Xinzhuang Zhang.

**Methodology:** Hui Zang, Jiankang Yu, Huibiao Jia.

**Project administration:** Jianping Zhou.

**Resources:** Jianping Zhou.

**Supervision:** Jianping Zhou.

**Validation:** Guanru Qian, Jingtong Tang.

**Writing – original draft:** Guanru Qian, Hui Zang.

**Writing – review & editing:** Guanru Qian, Hui Zang.

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
