## [Decision Letter · Decision Letter 0]

9 May 2024

Dear Mr. Zhou,

Thank you very much for submitting your manuscript "Lactobacillus gasseri ATCC33323 affects the intestinal mucosal barrier to ameliorate DSS-induced colitis through the NR1I3-mediated regulation of E-cadherin" for consideration at PLOS Pathogens. As with all papers reviewed by the journal, your manuscript was reviewed by members of the editorial board and by several independent reviewers. In light of the reviews (below this email), we are not in a position to accept this manuscript in its current format. We would consider a resubmission of a significantly-revised version that takes into account the reviewers' comments.

A number of significant issues concerning study design have been raised. In particular the lack of litter mate controls throughout the experiment. It is essential that experiments be performed in appropriate litter mate controls to account for alterations in the microbiome. In addition, the content of Lactobacillus gasseri present in the mice post transfer would need to be quantified. Should you choose to submit a revised version of this manuscript these issue would need to be addressed in full and new experimental data included.

We cannot make any decision about publication until we have seen the revised manuscript and your response to the reviewers' comments. Your revised manuscript is also likely to be sent to reviewers for further evaluation.

If you choose to resubmit, please upload the following:

Sincerely,

Rachel M McLoughlin, PhD

Academic Editor

PLOS Pathogens

Helena Boshoff

Section Editor

PLOS Pathogens

Michael Malim

Editor-in-Chief

PLOS Pathogens

orcid.org/0000-0002-7699-2064

A number of significant issues concerning study design have been raised. In particular the lack of litter mate controls thoughout the experiment. It is essential that experiments be performed in appropriate litter mate controls to control for alterations in the microbiome. In addition, the content of Lactobacillus gasseri present in the mice post transfer would need to be quantified. Should you choose to submit a revised version of this manuscript these issue would need to be addressed in full and new experimental data included.

Reviewer's Responses to Questions

**Part I - Summary**

Reviewer #1: Qian and colleagues studied the Lactobacillus gasseri ATCC33323 to identified mechanisms through which L. gasseri can modulate colitis. To this end they generated a transgenic E-Cadherin mice using the Cre/Lox-Villin system and expose the mice to dextran sodium sulphate to induce colitis and orally gavaged with L. gasseri. The authors show that L. gasseri reduced inflammation, improved barrier function, restored microbiota composition in wild type mice treated with DSS and L. gasseri but not in the E-cadherin transgenic mice. In addition, the authors also show that L. gasseri regulates E-cadherin potentially via the nuclear receptor NR1I3. Overall, the study presents some interesting findings on the interaction of L. gasseri and host via E-cadherin.

Overall, the study reveals an interesting set of data on L. gasseri and their health benefit associated with intestinal inflammation. The authors have used several techniques to support their conclusion, the number of animals per group is good for this type of study. Most of the figures are fine, some need some revision. The paper needs to be reviewed by an native English speaker and attention to detail for some contradictory sentences.

Reviewer #2: In this manuscript, the authors elucidate the role of Lactobacillus gasseri ATCC33323 in murine inflammatory bowel disease (IBD). According to the authors, Lactobacillus gasseri ATCC33323 mitigates physiological damage in colitis-afflicted mice by ameliorating the severity of colitis, reducing the production of inflammatory cytokines, and preserving the integrity of the intestinal epithelial structure and function. This protective effect is achieved through the modulation of CDH1 transcription by affecting NR1I3, thereby promoting the expression of E-cadherin. Overall, this work unveils a novel function of Lactobacillus gasseri ATCC33323 in inflammatory bowel disease. Despite considerable efforts, concerns about the mechanisms and study design persist.

Reviewer #3: The purpose of this study was to investigate the potential protective role of L. gasseri in mice in a model of colitis induced by DSS.

**Part II – Major Issues: Key Experiments Required for Acceptance**

Reviewer #1: - Lines 383-84- the authors mentioned a difference in mortality rate between the groups. Please replace Figure 1d, whose normalise data is presented in Figure 1e, with a survival curve to support the statement.

- Lines 393-95 please revise this statement as it doesn’t agree with the data in Figure 1f and the text in lines 395-98 “Similarly, the improvement in colon length was significantly reduced, and intestinal wall congestion was visible in the CDH1+/- LAB group”.

- Figure 4, the immunofluorescent stainings of b-catenin and p120 catenin are difficult to see and even more when they’re merged. Higher zoom might be helpful.

- Line 705-706 – the authors state “…E-cadherin semiknockout mice exhibit accelerated progression of colitis, which indicates that E-cadherin plays a crucial role in intestinal protection…”, which is not really in line with the data presented in this manuscript.

Reviewer #2: 1.The results indicate that Lactobacillus gasseri ATCC33323 can improve physiological damage in mice with colitis, but why does the title of result 2.1 (P357) suggest that Lactobacillus gasseri ATCC33323 causes physiological damage in DSS mice?

2.Why do results 2.1 and 2.2 have identical titles? Additionally, several titles do not adequately reflect the conclusions drawn.

3.What was the basis for selecting the following genes (IGF2, ATP2A1, PROX1, H2-OB, ESCO2, MKI67, LRR1, SYT8, BIRC5, PLA2G4E, UHRF1, EPHB3, NR1I3, and PBK), and subsequently, why was NR1I3 singled out? Given the results in Figure 7a-b, the change in NR1I3 is minimal, and the absence of statistical analysis of qPCR results, along with histograms lacking bars, calls the experimental conclusions into question.

4.The authors did not analyze fecal DNA from each mouse group to determine the content of Lactobacillus gasseri ATCC33323, failing to demonstrate the efficacy of the microbial transplant. The 16s RNA sequencing results also do not reflect an increase in lactic acid bacteria abundance.

Reviewer #3: While there is a lot of work in this study, it is not possible to interpret the data and their potential significance because of the study design. Indeed, based on the description of the study design, the two starting mouse lines are Cdh1flox/wt from a Cdh1flox/flox x Villin-Cre cross versus some control Cdh1 wt/wt mice (where were those mice coming from?). To allow interpretation, the control group should have been littermate mice of the Cdh1flox/flox group. In the absence of controlling for littermates, no conclusion can be reached because of the confounding effect of the microbiota.

**Part III – Minor Issues: Editorial and Data Presentation Modifications**

Reviewer #1: � Line 68-69 the authors state “…on probiotics, which have shown strong efficacy in the treatment of IBD[2]”. To date, the use of probiotics in IBD hasn’t been as successful as seen in preclinical models. Some efficacy studies have been reported in UC but not in CD. Please revise this strong statement and adapt according to current literature.

Line 81 – correct “zona encludens” with “zona occludens”

Line 89-90 – please revise the sentence “… these strains include ZO-1, occludin, claudin1, β-catenin, and E-cadherin”. I believe the authors are referring to junction proteins regulated by different Lactobacilli strains.

Lines 103-05 – what is the relevance of the 70 possible transcription factors identified in L. gasseri in context of this study or IBD? Please clarify or delete the sentence.

- NR1I3 is first time mentioned in line 180, introduce this factor in the introduction and why is relevant for the studies

Line 134 & 136 - please clarify if the group labelled as CDH1+/- LAB and NC LAB groups, are also exposed to DSS.

Please add references to the methodology section, where appropriate.

Line 210-11 …”the supernatant was collected from the eyeball blood”… please revise if you mean plasma or serum was collected.

Line 222 – information on the microscope is missing

Line 275 – which IL1 expression was assay e.g. IL1b?

Line 286 – what is the abbreviation CTAB?

Figure 1 c,d,e – please utilise the same colour for the same group to easily visualise the data for the reader

Figure 1b – provide ID on top of the 2 gels for clarity

Lines 401-03 revise this statement as no significant difference was observed between NC DSS and CDH1+/- DSS to their respective controls “The results illustrated that the diffusion of FITC-dextran in the intestine of mice was significantly increased after supplementation with DSS…”

Figure 2f, the PCR values are negative in the less inflamed and control animals, can you comment as to what was used as calibrator group in the calculations and include in the legend what the dotted line at zero stands for.

Refer in the text where appropriate to suppl figures 1 and 2 regarding the analysis of WB and staining of the representative figures 2-4.

Reviewer #2: 1.Please explain the rationale behind using colon cancer cells for in vivo experiments. Why not stimulate normal colon cells with pro-inflammatory factors to simulate the in vivo environment?

2.The western blots fail to display the molecular weight of each protein, casting doubt on the experimental conclusions.

Reviewer #3: (No Response)

PLOS authors have the option to publish the peer review history of their article (what does this mean?). If published, this will include your full peer review and any attached files.

Reviewer #1: No

Reviewer #2: No

Reviewer #3: No
---

## [Decision Letter · Decision Letter 1]

29 Aug 2024

Dear Mr. Zhou,

We are pleased to inform you that your manuscript 'Lactobacillus gasseri ATCC33323 affects the intestinal mucosal barrier to ameliorate DSS-induced colitis through the NR1I3-mediated regulation of E-cadherin' has been provisionally accepted for publication in PLOS Pathogens.

Best regards,

Rachel M McLoughlin, PhD

Academic Editor

PLOS Pathogens

Helena Boshoff

Section Editor

PLOS Pathogens

Michael Malim

Editor-in-Chief

PLOS Pathogens

orcid.org/0000-0002-7699-2064

Reviewer Comments (if any, and for reference):

Reviewer's Responses to Questions

**Part I - Summary**

Reviewer #1: (No Response)

**Part II – Major Issues: Key Experiments Required for Acceptance**

Reviewer #1: (No Response)

**Part III – Minor Issues: Editorial and Data Presentation Modifications**

Reviewer #1: (No Response)

PLOS authors have the option to publish the peer review history of their article (what does this mean?). If published, this will include your full peer review and any attached files.

Reviewer #1: No

---

## [Editor Report · Acceptance letter]

2 Sep 2024

Dear Mr. Zhou,

We are delighted to inform you that your manuscript, "Lactobacillus gasseri ATCC33323 affects the intestinal mucosal barrier to ameliorate DSS-induced colitis through the NR1I3-mediated regulation of E-cadherin," has been formally accepted for publication in PLOS Pathogens.

Best regards,

Michael Malim

Editor-in-Chief

PLOS Pathogens

orcid.org/0000-0002-7699-2064